# Application of CryoSat-2 altimetry data for river analysis and modelling

Raphael Schneider[1], Peter Nygaard Godiksen[2], Heidi Villadsen[3], Henrik Madsen[2], Peter Bauer-Gottwein[1]

[1]Technical University of Denmark, Department of Environmental Engineering, 2800 Kgs. Lyngby, Denmark
[2]DHI, 2970 Hørsholm, Denmark
[3]National Space Institute, Technical University of Denmark, 2800 Kgs. Lyngby, Denmark

*Correspondence to*: Raphael Schneider (rasch@env.dtu.dk)

**Abstract.** Availability of in situ river monitoring data, especially of data shared across boundaries, is decreasing, despite growing challenges for water resource management across the entire globe. This is especially valid for the case study of this work, the Brahmaputra basin in South Asia. Commonly, satellite altimeters are used in various ways to provide information about such river basins. Most missions provide virtual station time series of water levels at locations where their repeat orbits cross rivers. CryoSat-2 is equipped with a new type of altimeter, providing estimates of the actual ground location seen in the reflected signal. It also uses a drifting orbit, challenging conventional ways of processing altimetry data to river water levels and their incorporation in hydrologic-hydrodynamic models. However, CryoSat-2 altimetry data provides an unprecedentedly high spatial resolution. This paper suggests a procedure to i) filter CryoSat-2 observations over rivers to extract water level profiles along the river, and ii) use this information in combination with a hydrologic-hydrodynamic model to fit the simulated water levels at an accuracy that cannot be reached using information from globally available DEMs such as SRTM only. The filtering was done based on dynamic river masks extracted from Landsat imagery, providing high enough spatial and temporal resolution to map the braided river channels and their dynamic morphology. This allowed extraction of river water levels over previously unmonitored narrow stretches of the river. In the Assam Valley section of the Brahmaputra River, CryoSat-2 data and Envisat virtual station data were combined to calibrate cross sections in a 1D hydrodynamic model of the river. The hydrologic-hydrodynamic model setup and calibration are almost exclusively based on openly available remote sensing data and other global data sources, ensuring transferability of the developed methods. They provide an opportunity to achieve forecasts of both discharge and water levels in a poorly gauged river system.

## 1    Introduction and background

This study shows how river water level measurements from the drifting-orbit radar altimetry mission CryoSat-2 can be used in combination with hydrodynamic river models. This new type of satellite altimetry data, providing river water level profiles with unprecedented spatial resolution was used in combination with conventional data from Envisat, providing water level time series at virtual stations. The combination of these two datasets allowed accurate calibration of water level

dynamics – both absolute water levels as well as water level amplitudes – along a continuous stretch of a 1D hydrodynamic model of the Brahmaputra River. This is obtained without precise knowledge of topography or bathymetry.

## 1.1 Satellite altimetry over rivers

Satellite altimetry is often used in data scarce river basins such as the Brahmaputra basin. Numerous studies combining satellite altimetry with hydrologic river models have been carried out using data from repeat orbit satellites such as Envisat, ERS-2, TOPEX/Poseidon or Jason-1 and 2. Those satellites are on repeat orbits with a repeat cycle of 10 to 35 days (see for example Schwatke et al. (2015) for an overview of the main characteristics of current satellite altimetry missions). The laser altimeter mission ICESat, in operation from 2003 to 2009, has an unusually long repeat cycle of 91 days, resulting in a higher across track resolution. Processed ICESat data over inland waters though only became freely available recently (O'Loughlin et al., 2016). Repeat orbits simplify application in hydrologic studies, especially for rivers. First, observations occur only at a few locations along a river. This eases filtering of the data, as water masks which are commonly used to distinguish relevant points over the river from non-relevant points over land, only have to be applied to limited areas. Second, repeat orbits result in water level time series at certain points in the river – so-called virtual stations – a format commonly used in hydrology. Both does not apply to CryoSat-2: its drifting orbit results in water level measurements along the entire river, and the long repeat cycle of 369 days does not allow for direct derivation of water level time series (see the map inset in Figure 1 for a comparison of the Envisat and CryoSat-2 ground tracks). A good example for the focus of the hydrologic community on time series is the choice of Schwatke et al. (2015) to merge satellite altimetry from missions with differing orbits into common virtual stations for their satellite altimetry database DAHITI. Another effort to obtain a densified altimetry dataset is the work by Tourian et al. (2016): They merged multi-mission altimetry data over several rivers, linking the data between different virtual stations statistically and hydraulically, including data from CryoSat-2. The hydraulic link, however, is not a model but a simple time lag. They also found that the inclusion of CryoSat-2 data gives a more accurate representation of the river's water level profile. In our study, CryoSat-2 data were handled by filtering its Level 2 altimetry data over a dynamic river mask based on Landsat imagery and using the resulting spatially distributed data to calibrate the water level profile along a continuous stretch of a river model.

## 1.2 Combining satellite altimetry with river models

Many of the studies using satellite altimetry over rivers have been done for the Amazon River due to its large width and favourable direction of flow – predominantly west to east – in relation to altimetry satellite orbits (for example Yamazaki et al. 2012b and Paiva et al., 2013). Other examples include other big rivers, such as the Mekong and Ob in the work of Birkinshaw et al. (2014) where daily discharge data were estimated from Envisat and ERS-2 altimetry. A combination of MODIS data of river velocity and Envisat water levels was used by Tarpanelli et al. (2014) to estimate discharge in the Po River. Becker et al. (2014) used Envisat altimetry data to obtain a comprehensive dataset of water levels over the Congo basin, a poorly gauged river system. The dense sampling pattern of ICESat was used by O'Loughlin et al. (2013) to derive

water level slopes along the Congo River. ICESat river water levels were also used to evaluate the output of different hydraulic models (Jarihani et al., 2015 and Neal et al., 2012). Moreover, applications of data from the wide-swath drifting orbit mission Surface Water Ocean Topography (SWOT) have been considered (for example Biancamaria et al., 2011a or Yoon et al., 2012), however only with synthetically generated data: the SWOT mission is expected to be launched in 2020

(NASA, 2016). Calibration of hydrodynamic model parameters has been explored as well: Domeneghetti et al. (2014) calibrated channel roughness for a part of the Po River using multi-year Envisat and ERS-2 altimetry data. Their work relied on the availability of cross section surveys. A method using the 2D hydrodynamic model LISFLOOD-FP based on SRTM topography was suggested by Yan et al. (2014). They used Envisat altimetry to calibrate channel roughness and a parameter estimating channel bed elevation below the SRTM elevations representing the water surface. Similar is the work by

Biancamaria et al. (2009). They assumed fixed-width rectangular cross sections and estimated channel roughness and river depth by comparing model results to in situ discharge data and altimetry from Topex/POSEIDON. Cross section parameters were calibrated using ICESat altimetry in the lower Zambezi River (Schumann et al., 2013).

Also the chosen study area for this work, the Brahmaputra basin in South Asia, has already been used to show the value of Envisat altimetry data (Michailovsky et al., 2013). Furthermore, for example the work of Biancamaria et al. (2011b)

provided forecasts of water levels in the high-flow season for the Ganges and Brahmaputra River near the Bangladeshi border with the aid of TOPEX/Poseidon satellite altimetry. Based on the ideas from that study, Hossain et al. (2014) developed an operational flood forecasting system for Bangladesh using Jason-2 water level observations from the upstream parts of the Ganges and Brahmaputra River in India. Also basin water storage can be estimated from satellite altimetry, see the work of Papa et al. (2015) where a combination of Envisat altimetry and GRACE time-lapse gravimetry has been used to

estimate surface and sub-surface water storage in the Ganges-Brahmaputra basin.

Obviously, using satellite altimetry is particularly attractive over poorly gauged basins where in situ data are scarce. Satellite altimetry for river monitoring has recently become even more important: on the one hand, inland water altimetry is progressing with new satellites and sensors and improved data processing. On the other hand, despite growing challenges in managing our freshwater resources due to climate change, economic growth and population growth, the availability of in situ

river level or discharge data is decreasing in recent years. This can be seen amongst others in the amount of data that is archived in the Global Runoff Database (Global Runoff Data Center (GRDC), 2015). Robert Brakenridge et al. (2012) for example discuss that this is not only an issue of lacking in situ gauging stations, but often also a political decision to not share river monitoring data. In both cases, remote sensing data for example in the form of satellite altimetry as used in this work can help water resource management and flood prediction.

**1.3    Hydrodynamic river models**

If a river model is used to make predictions about water levels, a physically based discharge routing model has to be used. There exist 1D hydrodynamic models based on the Saint-Venant equations for unsteady flow, like the MIKE 11 model used in this study (Havnø et al., 1995). More complex 2D or coupled 1D-2D models also include the river flood plain and

interactions between channel and flood plain. The increased complexity of a 2D model compared to a 1D model obviously leads to higher computational demand. Furthermore, a meaningful setup of a 2D model requires more input data. Especially for large models in data scarce regions like the Brahmaputra basin, a compromise between computational efficiency and realistic simulation of water flow has to be made. Even though 2D models nowadays are successfully applied also to basin-

scale models (Biancamaria et al., 2009, Biancamaria et al., 2011a and Schumann et al., 2013) their computational demand still puts limits to the number of possible model runs (García-Pintado et al., 2013). For model calibration or data assimilation many model runs are required. Moreover, the setup of a 2D river model requires precise DEMs. Globally available datasets such as the SRTM DEM might not always be accurate enough, even if corrected specifically for their application in a river model (Yamazaki et al., 2012a). In such cases, a less complex 1D model is more robust.

**2    Study area**

The Brahmaputra basin in South Asia and its main river are being monitored closely by India and China, however almost none of this in situ hydrologic monitoring data are publicly available. The basin, for example, is considered a "classified basin" by the Indian government (Central Water Commission, 2009). This shows the importance of remote sensing data to aid any hydrologic modelling of the basin, such as flood forecasting in Bangladesh, the downstream neighbour of India.

Bangladesh, a low lying country at the Bay of Bengal in the estuary region of the three large rivers Ganges, Brahmaputra and Meghna is often hit by devastating floods. More than 90% of its surface water originates from outside the country, i.e. mainly India, but still little data are shared between Bangladesh and India (Biancamaria et al., 2011b). Because of the absence of trans-boundary data sharing, the region's dynamic hydrology, and the considerable size of the rivers in the Ganges-Brahmaputra-Meghna system, the area has repeatedly been in focus of river altimetry studies. These are also the

main reasons why the Brahmaputra basin has been chosen as a study area, despite making it hard to validate the altimetry data against in situ observations. As already mentioned, the Amazon River is another common study area for this kind of studies, but the river is exceptionally wide, making transferability of the applied methods hard.

Figure 1 shows a map of the hydrologic-hydrodynamic model of the entire Brahmaputra basin, which was set up in the DHI MIKE HYDRO River software (DHI, 2015). The course of the Brahmaputra River can be roughly divided into two parts: the

upstream part in the Tibetan Plateau and through the Himalaya into India, where the river is often flowing in steep valleys in a narrow river bed. River morphology changes in the downstream part as soon as the river leaves the Himalayan Mountains and enters the Assam Valley in India: here the Brahmaputra River is a wide, braided river with a low gradient and dynamically changing river channels (Sarkar et al., 2012). Finally, the Brahmaputra River merges with the Ganges and Meghna Rivers (outside the area modelled in this study) and flows into the Bay of Bengal.

Calibration of the hydrodynamic model's channel roughness was performed using discharge data from Bahadurabad station. River cross section calibration, however, proves more challenging as no accurate digital elevation model (DEM), river bathymetry, or other topographic information is available for the study area. Cross sections have to be calibrated along the

entire river, to allow realistic simulation of water levels. For this, a combination of conventional satellite altimetry from Envisat with the new data from CryoSat-2 was used. Envisat data, like other data from repeat orbit missions, enable derivation of water level time series at so-called virtual stations where the satellite ground track intersects the river. Cross section calibration had to be limited to the Assam Valley, as this is the only part of the river where sufficient data from both

CryoSat-2 and Envisat exist.

## 3    Data and methods

### 3.1    CryoSat-2 satellite altimetry data for rivers

In April 2010, the European Space Agency (ESA) launched CryoSat-2, a Synthetic Aperture Rader (SAR) satellite mainly designed to observe the cryosphere. However, it also proved useful to observe water levels over oceans and inland waters.

The data from CryoSat-2 are unique due to i) the satellite's drifting orbit and ii) its SAR Interferometric Radar Altimeter (SIRAL) sensor, making it possible to use a second antenna and then determine off-nadir positions of the radar reflections (European Space Agency and Mullar Space Science Laboratory, 2012). As will be shown in this work, this opens up for new applications of the data compared to conventional altimeters on a repeat orbit. CryoSat-2 is operating in three modes in different regions of the world determined by a geographical mode mask (European Space Agency 2016): In Low Resolution

Mode (LRM) as a conventional altimeter over the interior of ice sheets or regions of low interest to the cryosphere community; in SAR mode with an along-track footprint of only 300m (Wingham et al., 2006) for example over regions where sea ice is of interest; and in SAR Interferometric (SARIn) mode using a second antenna and other additional signal processing steps over areas with challenging terrain. Because of the two antennas used in SARIn mode, the signal's main reflectance location can be determined. This gives an estimate of the exact location of the measurement, instead of the

assumption that the measurement is placed directly at nadir as with conventional LRM and SAR altimetry data.

The data used for this study are Level 2 CryoSat-2 altimetry provided by the National Space Institute, Technical University of Denmark (DTU Space). These data were based on the ESA baseline-b Level 1b 20 Hz product, and retracked with an empirical retracker. For details of the processing please refer to Villadsen et al. (2015). Villadsen et al. also describe the application of the data over the Ganges and Brahmaputra rivers. Furthermore, Nielsen et al. (2015) were able to use these

data to extract water levels over lakes as small as 9 km$^2$ at unprecedented accuracy. Some of these data over lakes can be accessed via the Altimetry for inland Water (AltWater) service at http://altwater.dtu.space/ of DTU Space. For this work, data from CryoSat-2 are used from the beginning of its operation in 2010 until the end of 2013. Most of the Brahmaputra River is covered in SARIn mode, from its origin to the downstream end of the SARIn mask indicated in Figure 1, approximately 100 km upstream of the gauging station Bahadurabad.

### 3.1.1 Filtering of CryoSat-2 data – river mask

In the case of small inland water bodies, CryoSat-2 data, like any SAR altimeter data, currently do not deliver reliable information on whether it was acquired over water or over land surface. One relevant metadata item of satellite altimetry is the backscatter coefficient (also referred to as Sigma0). This value however, over small and often turbid water surfaces such as rivers does not allow a reliable discrimination between water and land surface points. In an effort of processing multi-mission data over inland waters, Schwatke et al. (2015) found backscatter only useful to deliver information about potential ice cover. This applies to both the Level 1b and Level 2 data. These challenges are also reflected in the processing of altimetry data developed for databases providing global inland water altimetry data that are described below.

Commonly, to filter relevant altimetry observations that represent river or lake surfaces, water masks derived from other remote sensing data are used. Existing global products include the MODIS river mask from the Moderate Resolution Imaging Spectroradiometer on board of the Terra and Aqua satellites. Those multi-spectral instruments can provide a mask with a high temporal resolution, however only at 250 m spatial resolution (Enjolras and Rodriguez, 2009). Often, those masks are also only used as static masks, see for example the MOD44W product (Carroll et al., 2009) used with CryoSat-2 data over the Brahmaputra and Ganges by Villadsen et al. (2015). Also the River&Lake dataset, an ESA project providing water level time series over inland water bodies globally from altimeters on board of ERS-2, Envisat and Jason-2, used a static water mask (Berry and Wheeler, 2009). Another database for satellite altimetry with global coverage, HydroWeb, uses simple rectangular masks at virtual stations where the satellites' repeat orbits intersect with the river (Rosmorduc, 2016), and then applies some outlier filtering based on single transects (Santos da Silva et al., 2010). Such a procedure cannot easily be applied to CryoSat-2 because of its drifting orbit. For the DAHITI database (Schwatke et al., 2015), which combines multi-mission data into common water level time series, a river mask is only applied via a simple latitude threshold (as all satellite tracks run in predominantly north-southerly direction). Then, further outlier criteria are applied to the data, including expected water height thresholds, height error thresholds, and along-track outlier tests. This procedure however also requires manual, individual inspection of river transects to tune the respective parameters.

The Brahmaputra in the Assam Valley has a braided river bed with river channels continuously changing their location and shape. Often, relevant changes can be seen from one year to another. Hence, for this work a high resolution, dynamic river mask was necessary for correct filtering of the CryoSat-2 altimetry data. This river mask was extracted from Landsat 7 and Landsat 8 NDVI imagery. Landsat imagery has been used repeatedly as a more finely resolved alternative to the global water masks discussed above, see for example the use with Envisat data over the Zambezi river by Michailovsky et al. (2012), with Envisat and ERS-2 data over the Mekong and Ob rivers by Birkinshaw et al. (2014), or the work by O'Loughlin et al. (2013) using river widths etc. extracted from Landsat NDVI imagery for a hydraulic characterisation of the Congo River.

32-day composites of Landsat 7 and 8 NDVI imagery, as available online from the EarthEngine (NASA Landsat Program, 2016) have been used to extract binary water masks over the entire Brahmaputra River covered in CryoSat-2's SARIn mode, where all areas with a NDVI value greater than zero were considered land surface, and the remaining parts water surface.

Because of optical imagery being unable to penetrate cloud cover, in this region it is not possible to acquire a complete river mask during each of these 32-day windows. The water mask extraction was done differently for the upstream and downstream portions of the river: upstream of the Assam Valley the Brahmaputra River bed is less dynamic, because the river is usually contained by a steep valley. For this part, available, i.e. cloud cover-free imagery from the years 2012 and
2013 was combined into one river mask. For the more dynamic Assam Valley, one river mask for each year was created from all available imagery for this year, resulting in four river masks for the relevant years 2010 to 2013. Only pixels that were water-covered in all usable 32-day composites of each year were considered water in the resulting mask. This means that the river masks represent an estimate of minimum water extent during each year. Manual inspection of Landsat imagery from different years has shown that most dramatic changes to the river's morphology become visible after each high-flow
season, i.e. each year in late autumn. Hence, from the beginning of each calendar year a new river mask was used.

### 3.1.2    Projecting CryoSat-2 data into model space

In order to filter CryoSat-2 data for use in combination with the 1D hydrodynamic model used in this work, the points of CryoSat-2 observations have to be projected onto the model river line.

This procedure is displayed in Figure 2. CryoSat-2 observations are first filtered over the river mask of the respective year,
and then projected onto their nearest neighbouring point on the model river line. The model river line is static over the entire simulation time. This is due to technical challenges in setting up and calibrating a model with a changing river line in MIKE HYDRO River. In addition, the hydrodynamic model's simulation results are relatively insensitive to the exact course of the river line as long as the 1D approximation adequately represents river conveyance. However, the model river line affects where the CryoSat-2 observations are mapped to the model.

### 3.2    Envisat virtual station data

The Envisat virtual station data to extract the water level amplitudes in the Brahmaputra River were taken from 13 virtual stations along the Assam Valley for the years 2002 to 2010. The data were taken from the River&Lake project database (Berry and Wheeler, 2009).

### 3.3    Hydrologic-hydrodynamic model

### 3.3.1    Hydrodynamic model

A model of the Brahmaputra Basin from the origin of the river to Bahadurabad station, close to the river's confluence with the Ganges River was set up in the DHI MIKE HYDRO River software. The Brahmaputra River was modelled over a length of overall 3090 km. See Figure 1 for an overview. River flow in MIKE HYDRO River (previously referred to as MIKE 11) is modelled using a 1D dynamic wave routing based on the Saint-Venant equations for unsteady flow (Havnø et al., 1995).
The governing equations are solved using a 6-point implicit finite difference scheme (Abbott and Ionescu, 1967). The

solution is computed on a staggered grid of alternating Q and h points. Simulated discharge is available at Q points only, while simulated water level is available only at h points. Cross sections can be placed anywhere along the river, but in the model they are always placed at h points. If necessary, cross section datums and shapes are linearly interpolated to achieve this. In our setup, the default distance between each Q and h point is 2.5 km The delineation of the river was based on the

SRTM DEM. However, it should be noted that a river line based on the relatively coarse SRTM DEM can deviate from the natural river's course (or its centre line), see Figure 2. Inaccuracies in the used DEM also explain the slight disagreements between the river's course, the location of the discharge station and the actual basin outline in the very flat part of the river valley around Bahadurabad station. Discharge routing in the 1D hydrodynamic model however is insensitive to the exact location of the river line. Furthermore, despite considerable changes to the river channel, the general discharge-water level

relationship in the Brahmaputra River seems to be fairly stable (Mirza, 2003). The hydrodynamic model was forced by simulated runoff from subcatchments. Applying this forcing, Manning's number was calibrated to a uniform value along the entire river (see also Figure 3) by minimizing RMSE between simulated and observed discharge at Bahadurabad station.

### 3.3.2    Rainfall-runoff forcing of the hydrodynamic model

Simulated runoff was derived from a larger hydrologic-hydrodynamic model of the Ganges and Brahmaputra Basins

developed  in a consultancy project of DHI and the International Centre for Integrated Mountain Development (ICIMOD). In this model, the runoff was simulated in 86 subcatchments (33 in the Brahmaputra Basin, indicated in Figure 1, and 53 in the Ganges Basin) using NAM rainfall-runoff models (Nielsen and Hansen, 1973). NAM is a lumped, conceptual rainfall-runoff model and can, as in this work, include snow melt processes. Due to restricted access to in situ data, the hydrologic model was based almost entirely on freely available remote sensing and other global data sources. For subcatchment delineation

and elevation zoning of the NAM snow melt module the SRTM DEM was used. Precipitation forcing was derived from TRMM v7 3B42 data (Tropical Rainfall Measurement Mission Project (TRMM), 2011). Temperature and evaporation forcings were derived from the ERA-Interim reanalysis products from the European Centre for Medium-Range Weather Forecasts (ECMWF) (Dee et al., 2011 and Berrisford et al., 2011).

In situ discharge data were available for only 11 of the 86 subcatchments. Most of these are located in the Nepalese

Himalaya and are part of the Ganges Basin. Only two of the available in situ discharge stations are located in the Brahmaputra Basin. Furthermore, in situ discharge data exist for both main rivers, the Ganges and Brahmaputra River, close to their confluence with each other, at Hardinge Bridge station and Bahadurabad station respectively. The existing subcatchment in situ discharge was used to calibrate parameters controlling the size of the surface and root zone storage, thresholds between overland runoff, interflow and groundwater recharge, and time constants for overland flow, interflow and

baseflow routing. For details please refer to the reference manual, chapter 4.8 (DHI, 2015). Parameters for the 11 subcatchments were calibrated individually and then transferred to the remaining subcatchments, using simple heuristics. Parameters in the NAM model have some physical meaning, so for example differences in topography or land use can guide in how to transfer parameters from one catchment to another. So, even though only the Brahmaputra basin was part of this

study, information on NAM parameters gained from the larger Ganges-Brahmaputra model was used. The overall performance evaluation of the model, and a calibration of the Manning's number of the hydrodynamic model was done by evaluating the model output at Bahadurabad station at the outlet. Aggregated information at the outlet however could not directly provide information about subcatchment parameters. The calibration period included the years 2002 to 2007.

### 3.3.3 Boundary and initial conditions of the model

The hydrodynamic model was initialized from arbitrary initial conditions and warmed up for a sufficiently long period prior to the start of the actual simulation period. More relevant, however, are the initial conditions of the hydrologic model: some of the NAM model storages, such as groundwater and snow storage, have long residence times. Hence, the NAM models have been run for 30 iterations of the calibration period, until model states reached equilibrium. The resulting model states were then used as initial conditions for the simulation period. Furthermore, water level data from Aricha available for the years 2001 to 2009 was used as a downstream boundary condition of the hydrodynamic model. Outside this period, a daily water level climatology derived from the available observations has been used. Due to the location of Aricha approximately 180 km downstream of Bahadurabad station this has a negligible effect on results.

### 3.4 Cross section calibration

In the absence of in situ or precise local remote sensing data of the Brahmaputra River's cross sections or bathymetry, we used the SRTM DEM to derive the river's course with DEM hydroprocessing routines. The SRTM DEM, in combination with satellite imagery, was also used for a first guess of cross section datums and shapes along the Brahmaputra. These cross sections are a result of the DHI/ICIMOD project. Already here, due to the use of a 1D hydrodynamic model, the multi-channel river was simplified into one channel only. Due to the low spatial resolution of the SRTM DEM (90 metres) and the vertical standard error in the range of a few metres (Rodríguez et al., 2006), this provides a rough estimate of cross section datum. Furthermore, the SRTM DEM does not retrieve the submerged part of the river cross section, which has to be estimated or guessed. Hence, a hydrodynamic model with cross sectional data derived from such a DEM cannot be expected to accurately simulate water levels. As CryoSat-2 observations will occur along the entire river, and not only limited to virtual station locations, water levels have to be reproduced accurately by the model along the entire river, if the (or any) altimetry data were to be combined with the model.

The cross section calibration described in the following, fitting simulated water levels to observed water levels from altimetry, could only be performed for the downstream part of the Brahmaputra River, the Assam Valley. This is due to insufficient altimetry data from CryoSat-2 and Envisat available for the upstream part of the Brahmaputra River. Cross section calibration was performed after calibration of the Manning's number as described in the previous section.

For the hydrodynamic model, conceptual cross sections in triangular shape were placed at regular 50 km or 12.5 km intervals along the river; with information from the SRTM DEM as a first guess. This generic, simple shape was chosen to ease the calibration process. Cross section parameters (datum and opening angle) were then calibrated using information obtained

from both satellite altimeter data sources mentioned above to fit i) the average absolute water level along the river and ii) the water level amplitudes in the river. In combination, the two altimeter missions proved to provide very useful insight into all relevant dynamics of water levels in the river (profiles of average absolute water levels and water level amplitudes) due to their different orbits.

5 Figure 3 summarises the entire calibration process of the hydrologic-hydrodynamic model. It is assumed that the cross section calibration has negligible influence on (the timing of) discharge. With the used hydrodynamic model this holds true if a reasonable first guess for the cross sections was made.

### 3.4.1    Step 1: cross section calibration using average water levels

The drifting orbit of CryoSat-2 allows derivation of average water level profiles along the river with high spatial resolution if 10 several years of data are taken into account. These water level profiles are of higher accuracy than what can be extracted from SRTM DEM because of the lower standard error of CryoSat-2 altimetry data. Besides that, the CryoSat-2 data are filtered specifically to include only data of the actual water surface, unlike the SRTM DEM that averages different landcover and terrain types within one 90 m pixel. Also the dynamic river morphology requires use of current water level observations, instead of historic SRTM data which was acquired in 2000. This means that in the first step, as displayed in Figure 4, the 15 cross sections' datums were calibrated to fit the average simulated water level profile along the river to the average water level profile observed by CryoSat-2.

### 3.4.2    Step 2: cross section calibration using water level amplitudes

In the second step, the Envisat virtual station water level time series were used to calibrate cross section shapes to fit the water level amplitudes at the locations of the 13 virtual stations (see Figure 1) along the Assam Valley. The information used 20 here was the relative water levels, i.e. simulated yearly water level amplitudes were fitted to the observed ones from Envisat. To account for the coarse temporal resolution of Envisat data of 35 days and the resulting risk of losing a peak, simulated data were only extracted at the exact times of Envisat observations to determine the simulated water level amplitudes.

To perform the calibration, the MIKE HYDRO River model was coupled with a genetic search algorithm implemented in Matlab for numerical optimisation. Table 1 gives an overview over calibration parameters and objective functions used for 25 both steps.

As cross sections were placed every 12.5 km or 50 km, i.e. at finer intervals than virtual station observations were available, cross section angles were interpolated linearly for cross sections without any neighbouring virtual station. By doing so, 27 cross sections (Table 1) could be calibrated with information from 13 virtual stations. The change of the cross section shape in calibration step 2 has a relevant effect not only on the water level amplitudes but, at some points in the model, also on the 30 absolute average water levels. Consequently, step 1 of the water level calibration procedure has to be repeated with the cross section shapes that resulted from step 2. This leads to an iterative process displayed in Figure 3. It usually can be ended after few iterations at step 1, as simulated water level amplitudes are insensitive to moderate changes in the cross section datums.

# 4 Results and Discussion

## 4.1 Water level from CryoSat-2 data

The filtering of CryoSat-2 data over the Landsat river masks for 2010 to 2013 resulted in 4806 single observations. A

filtering of obvious outliers was performed, excluding CryoSat-2 values which deviate more than 20 metres from the SRTM elevations along the model river line. After outlier filtering, 3868 CryoSat-2 observations remain. Figure 5 displays longitudinal profiles of outliers and outlier-filtered values, as well as mapped outlier-filtered values along the course of the Brahmaputra River. The value of 20 metres was chosen after inspection of the data. It ensures removal of all obvious outliers, maybe due to issues with the closed-loop control of CryoSat-2. 2710 of the outlier-filtered measurements lie in the

Assam Valley of the Brahmaputra River, which were used for the cross section calibration. The number of data points and outliers are summarized in Table 2.

In the upstream part of the Brahmaputra, from river km 0 to approximately km 2100, there is a considerable amount of outliers. This is the part characterised by a steep or even gorge-like river valley (Jain et al., 2007). All the CryoSat-2 data used here are acquired in SARIn mode which allows determining the true ground location of the observation. However, the

applied off-nadir ranging is also connected with uncertainties (Armitage and Davidson, 2014). This could be one reason for the large amount of outliers in this part of the river. Furthermore, these outliers – most of them are clear outliers, with elevations hundreds of metres off – can be related to the steep valley making it impossible for the altimeter's signals to reach the valley bottom, i.e. the river water surface. Note in this context that CryoSat-2's measurement footprint area is 0.5 km$^2$ (Scagliola, 2013) with an along-track resolution of about 300 metres (European Space Agency and Mullar Space Science

Laboratory, 2012) in SARIn mode. Envisat, for example, has a measurement footprint diameter of 2 to 10 km (Chelton et al., 2001) and an along-track resolution of 369 metres (Berry et al., 2008), making it more likely for CryoSat-2, especially over challenging terrain, to lock onto the target of interest. Another issue with steep terrain is that the range window, in which an altimeter actually records potential reflections from the surface, constantly has to be adjusted according to the terrain. CryoSat-2's closed-loop control in SARIn mode means that it often misses to reach valley bottoms in mountainous regions

(Dehecq et al., 2013). It can be seen in the graph in Figure 5 that the steepest parts of the river (around river km 500, 900, 1250, and between km 1600 and 2000) contain almost no usable data at all. These steepest parts of the river often coincide with the most narrow parts of the river valley. The Assam Valley however requires almost no outlier filtering at all. Furthermore, it can be seen that CryoSat-2 captures some details in the river bed level that the SRTM data are not showing – see for example the detail around river km 1200 in Figure 5. And the CryoSat-2 data give an idea of the seasonal variability

of water levels along the river.

It can be concluded that CryoSat-2 altimetry data can be used over the parts of the Brahmaputra River with moderate topography. In the areas with extreme topography, a large amount of outliers can be found, however still leaving a relevant

amount of usable observations. This is important as none of the existing inland water altimetry databases (River&Lake, HydroWeb, DAHITI) provides data over the upstream part of the Brahmaputra River.

The use of a better river mask would likely improve the data yield. Such river masks should have a high temporal resolution (at least seasonal), and a high spatial resolution (well below the widths of the river channels of a few hundred metres). Using SAR imagery should give better results than the optical Landsat imagery used here: SAR imagery penetrates cloud cover, also giving results in the high-flow season of the Brahmaputra River where it was never possible to get consistent optical imagery. Since the start of Sentinel-1A as part of the Copernicus programme in April 2014, a freely available source for high resolution SAR imagery exists (Sentinel-1 Team, 2013) that could largely improve the river masks used for this work.

## 4.2 Hydrologic-hydrodynamic model calibration of discharge

After the calibration of the rainfall-runoff models the hydrodynamic model was calibrated to in situ observations at its outlet, Bahadurabad station (see Figure 1). This was done by adjusting Manning's number, affecting the timing of the discharge routing. The optimal Manning's number was found to be 0.029 $s/m^{1/3}$, which is considered plausible (compare Chow (1959), Table 5-6). Furthermore it was observed, both for the single catchments and the entire Brahmaputra River at Bahadurabad, that the precipitation forcing is likely underestimating the real precipitation. It was necessary to scale the TRMM precipitation data with a factor of 1.4 to obtain a good water balance. An underestimation of precipitation by remote sensing data can be observed sometimes, especially in regions with a large share of small-scale convective rainfall events. Also Michailovsky et al. (2013) observed in their work that the TRMM 3B42 had to be scaled by a factor of 1.25 to give good results in a hydrologic model of the Brahmaputra Basin. Moreover, given the large size of the subcatchments, spatial variation of precipitation due to topography (which is present in the Himalaya, see for example Bookhagen & Burbank, 2006) cannot be fully accounted for. Figure 6 shows simulated and observed discharge at Bahadurabad station.

Table 3 gives an overview over performance criteria, comparing observed and simulated discharge at Bahadurabad station for the calibration and validation period. For the validation period 2010 to 2013 data were usually only available during the high-flow season April to October. Hence, for comparability Table 3 also lists values for the calibration period taking only April to October into account. The good performance for the calibration period with a Nash-Sutcliffe coefficient (NSE) of 0.93, or 0.89 respectively, is reduced for the validation period to a NSE of 0.81. Also, the low water balance bias of around -2% in the calibration period increases to +11% for the validation period, meaning that the model is overestimating the discharge at Bahadurabad station. It is noticeable that the rainfall-runoff model performance on the subcatchment level is poorer than the performance of the aggregated model at the basin outlet (see Table 4). For subcatchments with available discharge observations the average NSE is 0.47. The poor subcatchment-level performance can maybe be explained by small scale precipitation patterns not properly represented by the TRMM precipitation product. The errors on the subcatchment level are almost uncorrelated to each other: the average Pearson coefficient of cross correlation of the subcatchments' runoff residuals is only 0.16. When these runoffs then are aggregated in the hydrodynamic model the uncertainties from the

individual subcatchments cancel each other out to some degree, leading to a much better performance on the basin level than on the subcatchment level. This can also be seen as an indicator that the NAM model parameter transfer was successful.

## 4.3    Cross section calibration

Figure 7 shows the results of step 1 of the water level calibration. For better visibility, the results are all shown in elevations relative to the reference model's cross section datums instead of absolute elevations. The reference model was run with the first-guess cross sections derived from the SRTM DEM described in section 3.4. It can be seen that the average simulated water levels from the reference model do not accurately represent the CryoSat-2 observations. After calibrating the cross section datums – which meant adjusting their datum by up to 4 metres – the simulated average water level follows the CryoSat-2 observations more closely. The calibration reduced the RMSE between average simulated water level and CryoSat-2 observations from 3.1 metres for the reference model to 2.5 metres. The remaining deviation can mainly be explained by the seasonal water level variations in the river.

While studying first results from this calibration step, we realized that between river km 2050 and 2150 the river bed slope is changing multiple times and finer cross section spacing is needed to accurately represent river morphology. Hence, in this part additional cross sections were added reducing the cross section spacing from 50 km to 12.5 km.

For one of the virtual stations, the results of step 2 of the cross section calibration can be seen in Figure 8. The average RMSE between simulated and observed yearly water level amplitudes for all 13 virtual stations was 0.83 metres after the calibration.

It has to be noted that CryoSat-2 data was used approximately from river km 1950 to 2800, whilst Envisat data was available from river km 2050 to 3050. Hence, only the overlapping stretch from river km 2050 to 2800, spanning a total of 22 cross sections in the chosen setup, can be considered fully calibrated.

The cross section calibration procedure developed offers a way to obtain a rather simple 1D hydrodynamic model accurately representing water levels, without precise knowledge of topography or bathymetry. Synthetic cross sections allow the use of practically any shape, however for the sake of reducing the number of fitting parameters in the calibration algorithm a simple triangular shape was chosen. These simple cross section shapes proved to be able to reproduce the observed water level amplitudes. Also, other physical properties of the hydrodynamic model are in a plausible range: For the calibrated stretch of the Brahmaputra River, the average Froude number at the model's individual grid points varies from 0.086 to 0.4149 in the high flow season, and from 0.070 to 0.399 in the low flow season. The average simulated water depth varies from 1.80 to 9.77 metres in the high flow season, and from 1.02 to 6.08 metres in the low flow season. This means that the modelled flow is well in the subcritical range, as expected for the given river section. Still it has to be stressed that properties other than discharge and water level will not be represented realistically. Trigg et al. (2009) had similar success with simplistic cross section geometries: They introduced only marginal errors in water levels from a hydrodynamic model of the Amazon River when switching from surveyed cross sections to rectangular representations. However, for some rivers, it might be impossible to model the observed discharge-water level relationships with such simplistic cross sections. The approach can

be adapted for a slightly more complex representation of cross section geometry, as for example suggested by Neal et al. (2015).

## 5 Conclusion

This is one of the first studies demonstrating how to use Cryosat-2 type radar altimetry data in connection with river models.

There have been other suggestions on how to use spatially distributed satellite altimetry in combination with hydrologic models. Often, however, they still rely on the concept of virtual stations; see the review in the introduction section. Other studies fall back on data from in situ gauging stations such as the work by Getirana (2010) using Envisat data to calibrate a model of the Negro River in the Amazon Basin, where data from a network of in situ gauging stations was used to estimate the water level-discharge relationships in a hydrologic model.

The method developed in this study, combining altimetry data from two missions with different orbits with a hydrologic-hydrodynamic model allows the calibration of cross sections in a 1D hydrodynamic river model without precise knowledge of topography or bathymetry. This results in a model accurately simulating water levels, which is an important achievement if poorly gauged river basins are to be modelled. Globally available DEMs such as the SRTM product are used to create hydrodynamic models, though they do not always provide enough information to reproduce water levels or inundations areas

at high accuracy. Jarihani et al. (2015) used cross sections derived from the SRTM DEM and different hydraulically and vegetation corrected versions of it, and compared to elevations from ICESat and surveyed points. Even correcting the SRTM for vegetation and submerged parts, a relevant error remained. Similar work was done by Md Ali et al. (2015), who compared water levels in a hydrodynamic model based on the SRTM DEM with those from a model based on more accurate lidar data. The resulting simulated water levels showed relevant differences. Similar conclusions can be drawn from this

study.

Envisat data – similar to data from other conventional altimeters such as ERS-2, Jason-2 or TOPEX/Poseidon – were used as a virtual station time series to extract water level time series. This kind of water level data are directly accessible from inland water satellite altimetry databases such as River&Lake, HydroWeb, or DAHITI. DAHITI started incorporating CryoSat-2 data in their multi-mission product. Still, CryoSat-2 data as river water levels currently cannot be accessed directly through

any of those sources. Hence, for this work a filtering procedure based on dynamic Landsat river masks was developed and applied to the CryoSat-2 data. This procedure took the dynamic nature of the Brahmaputra River's morphology into account and allowed to extract river water levels also over narrow parts of the river in extreme terrains, where existing global databases of inland altimetry do not offer any data. These data cannot be used to directly extract water level time series, but they display longitudinal water level profiles.

The hydrologic-hydrodynamic model of Brahmaputra River Basin that was used in conjunction with the satellite data, has been set up almost exclusively using openly accessible remote sensing and other global data sources. Thus, the methodology developed is transferrable to other case studies. The applicability of the suggested cross section calibration is only limited by

the availability of a sufficient amount of (satellite) altimetry observations, which mainly depends on river width and topography. The resulting calibrated model can be used for operational river discharge or water level forecasting, and be further informed by assimilating discharge measurements or water level measurements from various sources such as different altimetry missions.

## 5 Acknowledgements

The work presented in this paper was partly funded by the LOTUS – Preparing Sentinel-3 SAR Altimetry Processing for Ocean and Land project, part of the European Union's Seventh Framework Programme for research, technological development and demonstration under grant agreement no. 313238; and the ESA Cryosat2 Success over Inland Water and Land (CRUCIAL) project, awarded under ITT ESRIN/AO/1-6827/11/I-NB. The setup and calibration of the Brahmaputra model used in this study was carried out in collaboration with the International Centre for Integrated Mountain Development (ICIMOD).

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

**Figures and tables**

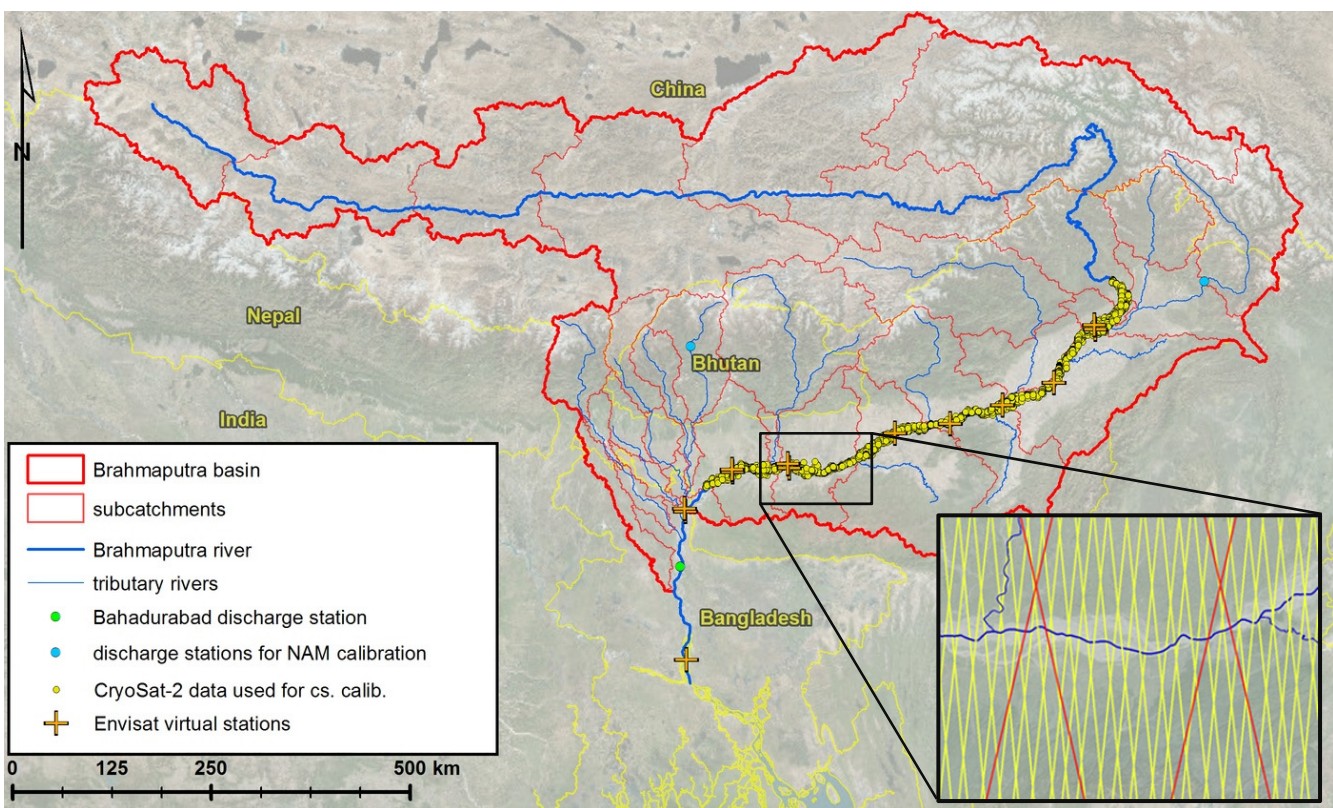

Figure 1: Brahmaputra basin model base map, showing the altimetry data used for the water level calibration (the entire upstream part of the Brahmaputra is also covered by the SARIn mode of CryoSat-2). Map inset: Ground tracks of CryoSat-2 (yellow) and Envisat (red). The Brahmaputra River originates in the Tibetean Plateau, flows eastwards, then bends towards south to cross the Himalayan Mountains and afterwards flows westwards through the Assam Valley into Bangladesh.

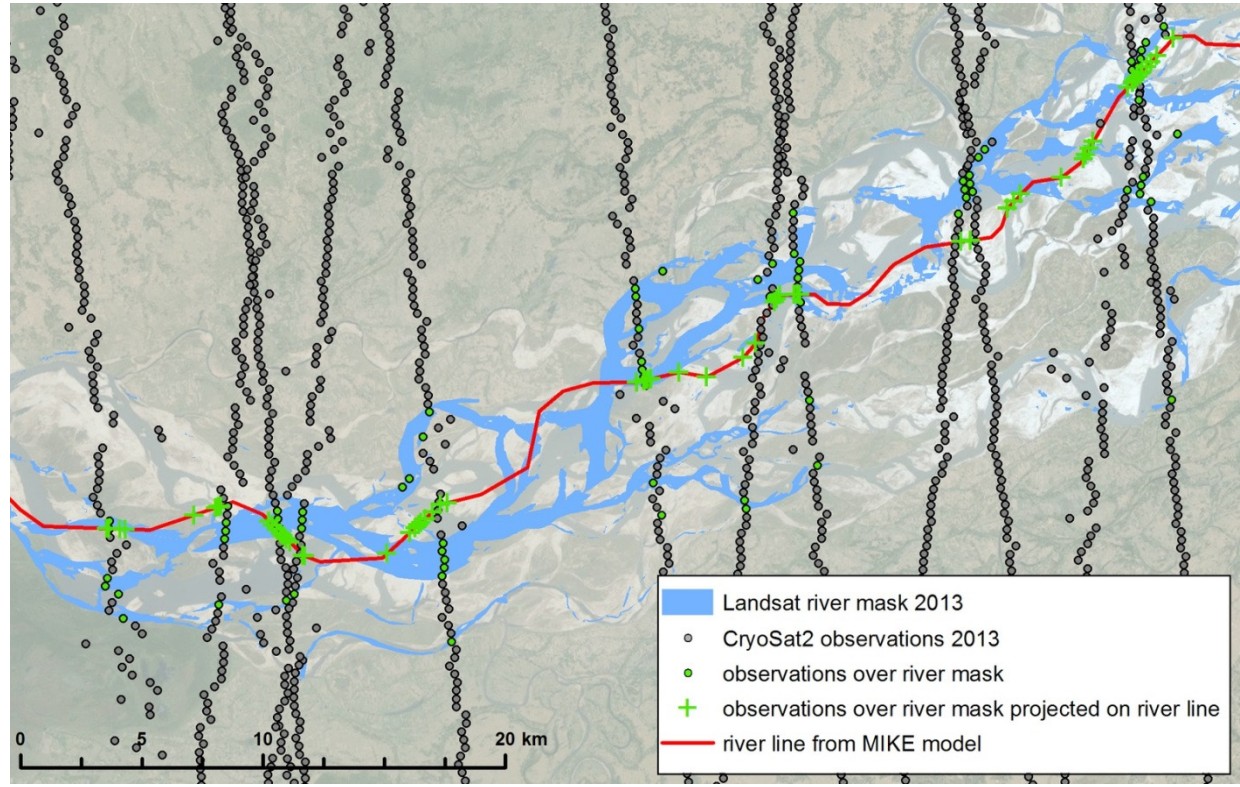

**Figure 2: Section of the Brahmaputra in the Assam Valley showing the Landsat river mask, the CryoSat-2 observations and their mapping to the 1D river model, all for 2013**

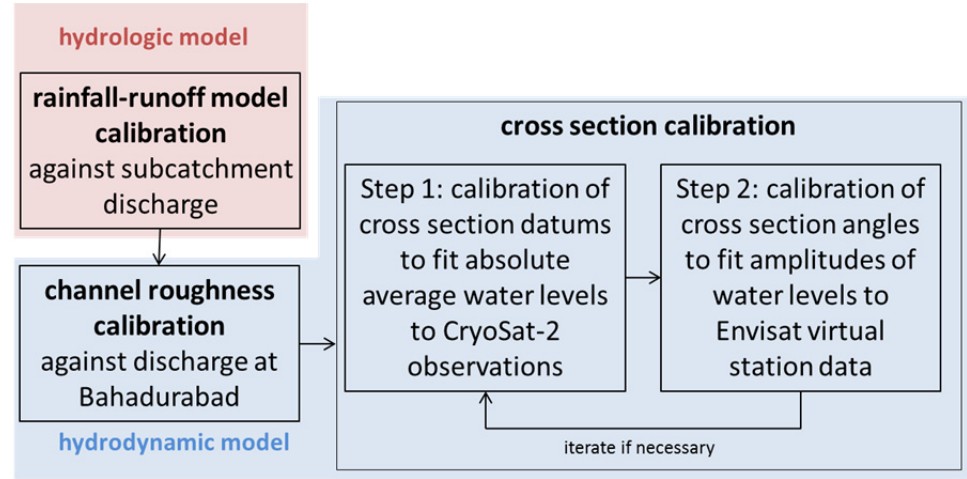

**Figure 3: Flow chart showing the hydrological and hydrodynamic model calibration.**

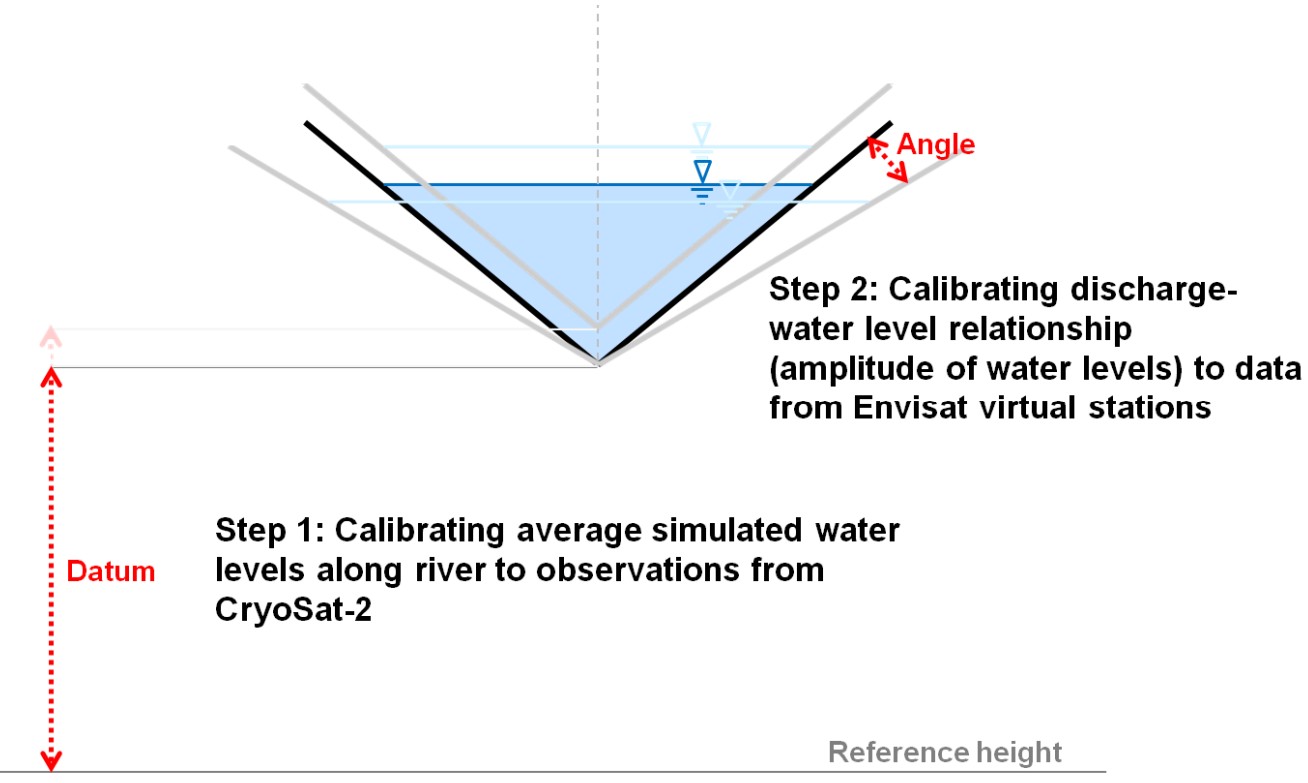

**Figure 4: Sketch of the two-step cross section calibration with their calibration parameters.**

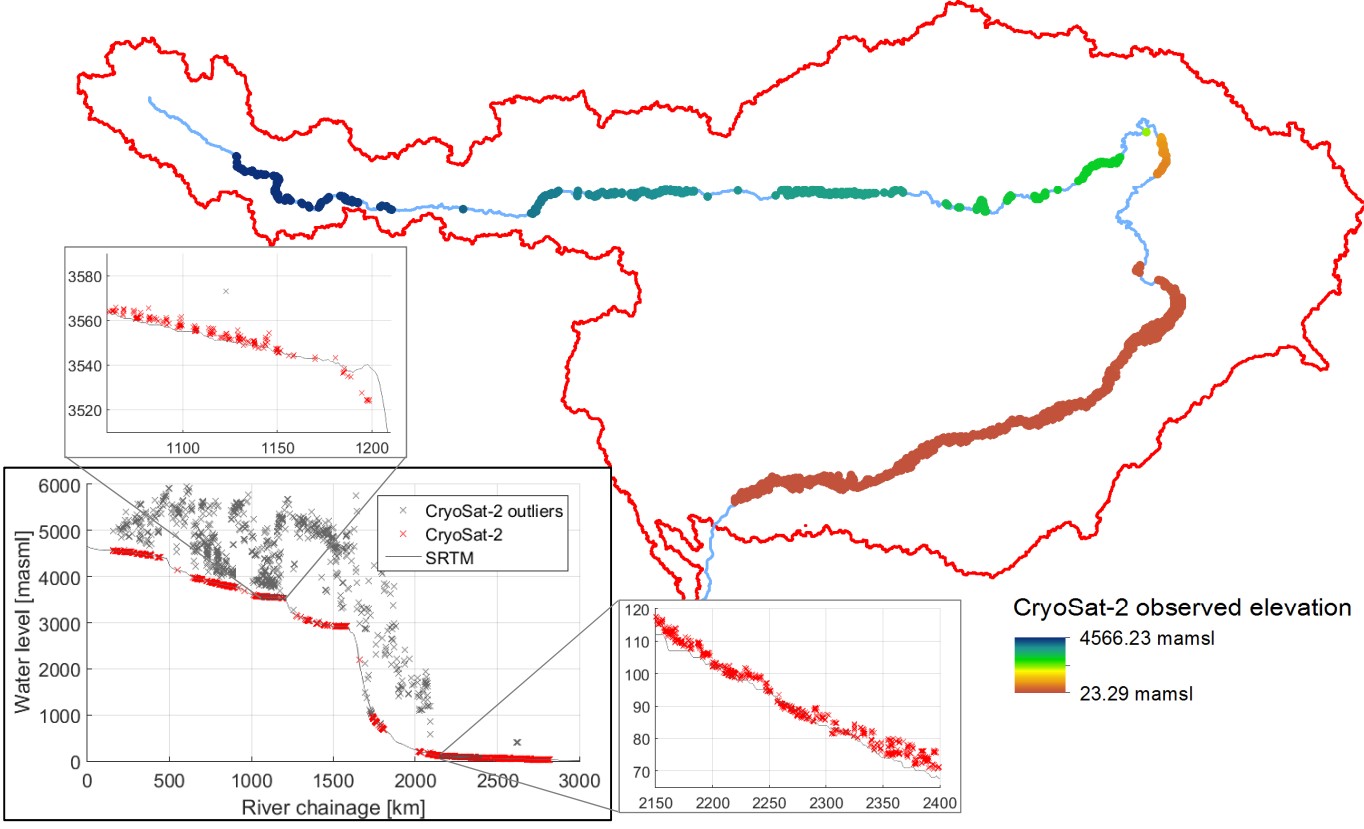

**Figure 5: CryoSat-2 observations along the Brahmaputra River from 2010 to 2013. The map only displays the outlier-filtered observations, the longitudinal profiles show both outliers and the outlier-filtered data.**

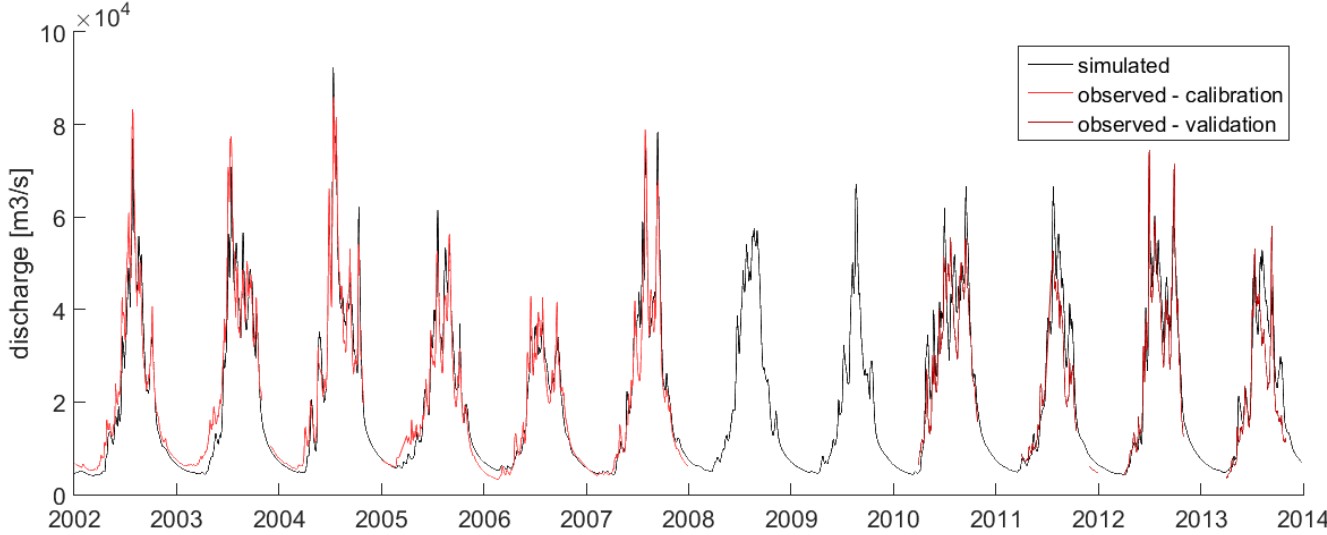

**Figure 6: Observed vs. simulated discharge from the hydrologic-hydrodynamic model at Bahadurabad station. 2002 – 2007: calibration period. 2010 – 2013: validation period.**

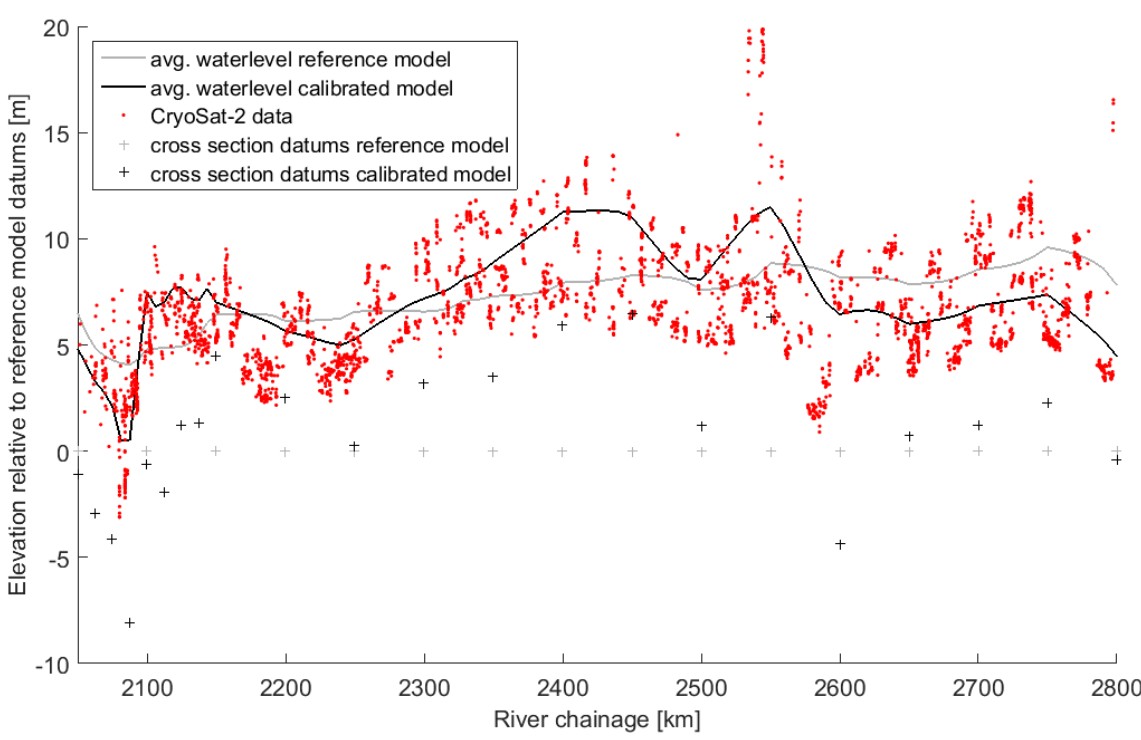

**Figure 7: Result of cross section calibration step 1 for the Assam Valley for the period 2010 to 2013. All levels are shown relative to the reference model's cross section datums based on the SRTM DEM.**

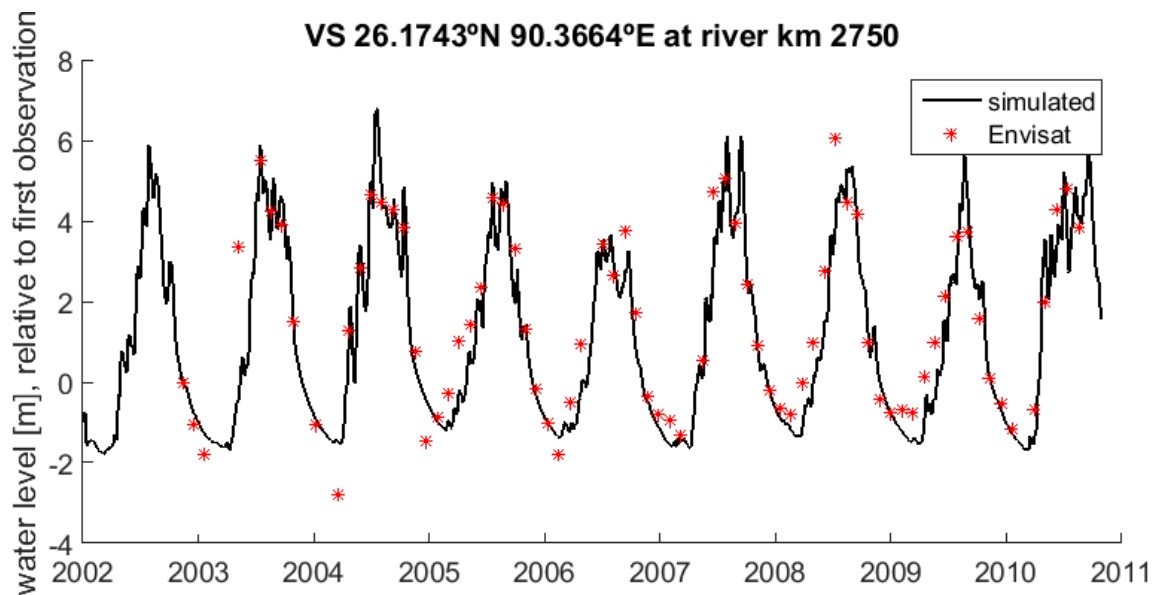

**Figure 8: Result of cross section calibration step 2 for one virtual station. All levels are relative to the water levels at the time of the first Envisat observation**

Table 1: Decision variables, constraints, and objective functions of the genetic algorithm used for the two-step cross-section calibration

|  | Calibration parameters | Constraints | Objective function |
|---|---|---|---|
| **Step 1: fitting absolute average water levels to CryoSat-2 observations** | 24 cross section datums, from river km 1950 to 2800 | Cross section datums continuously decreasing from upstream to downstream | RMSE between CryoSat-2 observations and average simulated water levels from 2010 to 2013 |
| **Step 2: fitting amplitude of water levels to Envisat virtual station data** | 27 cross section angles, from river km 2050 to 1800 | Cross section datums without neighbouring virtual stations linearly interpolated from their neighbours | RMSE between yearly amplitudes of Envisat virtual station data and of simulated water levels from 2002 to 2010 |

Table 2: Number of CryoSat-2 observations and outliers over the river mask of the Brahmaputra River from 2010 to 2013. River km 2100 and downstream is also referred to as Assam Valley.

|  | entire Brahmaputra | river km 0 – 2100 | river km 2100 – 2820 |
|---|---|---|---|
| **all CryoSat-2 observations** | 4806 | 2092 | 2714 |
| **outliers** | 938 | 934 | 4 |
| **filtered CryoSat-2 observations** | 3868 | 1158 | 2710 |

**Table 3: Performance criteria for simulated discharge $Q_{sim}$ at Bahadurabad station. Bias is given as $(Q_{sim} - Q_{obs})/Q_{sim}$.**

|  | RMSE [m³/s] | NSE [-] | bias [%] |
|---|---|---|---|
| **Calibration period 2002 – 2007** | 4329 | 0.93 | -2.1 |
| **Calibration period, high-flow only** | 5323 | 0.89 | -2.3 |
| **Validation period 2010 – 2013** | 6873 | 0.81 | 11.2 |

**Table 4: Performance criteria for simulated discharge in the calibration subcatchments. 2002 - 2007. \* indicates subcatchments in the Brahmaputra basin.**

|  | $Q_{sim}$, mean | $Q_{obs}$, mean | RMSE [m³/s] | NSE [-] |
|---|---|---|---|---|
| Arun | 437 | 529 | 259 | 0.65 |
| Bagmati | 254 | 110 | 364 | -0.18 |
| Bheri | 252 | 307 | 115 | 0.88 |
| Gandhak 1 | 720 | 955 | 535 | 0.80 |
| Kaligandaki | 236 | 403 | 264 | 0.69 |
| Karnali | 398 | 513 | 281 | 0.68 |
| Lohit* | 453 | 919 | 1045 | 0.03 |
| Rapti 1 | 241 | 115 | 212 | 0.12 |
| Sankosh 1* | 184 | 348 | 246 | 0.40 |
| Sunkoshi | 517 | 745 | 426 | 0.75 |
| Tamor | 201 | 416 | 392 | 0.39 |
| **mean** | 354 | 487 | 376 | 0.47 |

