# Peer review of "Application of CryoSat-2 altimetry data for river analysis and modelling"

_Hydrology and Earth System Sciences, 2016_

## Referee Comment (RC1) · Anonymous Referee #1 · 18 Jul 2016

GENERAL COMMENTS:

The paper is interesting because it shows a practical use of Cryosat-2 data for a hydrodynamic modelling. So far, a few studies are available on this issue in the scientific literature. Therefore, I found the paper highly timely and appealing.

The manuscript is well written and easy to follow, even if some aspects should be better clarified. The main issues concern: 1) the specification of the paper purpose, 2) the description of the hydrodynamic model and 3) the procedure of optimization of the cross-section geometry. Moreover, I have doubts concerning the study area characteristics. The evaluation of the Cryosat-2 data performances cannot be exhaustively tackled if no data are available for the validation.

SPECIFIC COMMENTS:

[Figure]

Introduction:

1) The purpose of the study is not well specified. I suggest the authors to add in the introduction a couple of sentences on this aspect also to introduce the model and the datasets used: why do they use 1D model for this complex river? Why software MIKE 11? Why Cryosat-2 and Envisat?

2) I believe that the background should be addressed following the purpose of the paper. The literature review described in the introduction is quite extensive, but it should be more focused on the use of radar altimetry for the calibration of the hydrodynamic models or the cross-sections geometry, mentioning similar studies (see references). For example:

Domeneghetti et al. (2014; 2015) compared the performances and analyzed the uncertainty of ERS-2 and ENVISAT radar altimetry in the calibration of the manning coefficient of the Hec-RAS model along a river reach of the Po river in Italy.

Yan et al. (2014) calibrated the manning roughness coefficient and the depth of the cross sections for the LISFLOOD-FP model in the Danube River with the use of water surface level derived by Envisat radar altimetry.

Biancamaria et al. (2009) compared the water levels derived by 22 TOPEX/POSEIDON VSs with the ones simulated by large scales coupled hydrological-hydraulic model of the Ob river in Siberia calibrating the river depth and Manning' roughness coefficient.

3) I suggest citing Tourian et al. (2016) for the merging of satellite altimetry. They analyzed different time series from Envisat, Saral/Altika, Topex/Poseidon and Cryosat-2 in the Po, Congo, Mississippi and Danube rivers.

Study area:

1) Why do the authors focus on Brahmaputra River? Cryosat-2 data are available for rivers where the in-situ data could be easily obtained. The risk to use a poorly gauged river (or as in this case a river where the data are not publicly available) is to be not

able to validate the procedure in a proper manner.

2) I have doubts on the use of "calibration" term in the text: "discharge calibration" or "water level calibration". The calibration is referred to the parameters of the model in order to reproduce the measured discharge or water level. I guess that, in this case, the authors calibrate the parameters of the hydrodynamic model and, then, compare the simulated discharge with the observed one. Therefore, I suggest to pay attention.

Data and Methods:

1) This section is quite unbalanced. The description of the satellite data, especially for the water mask, is too long with respect to the hydrodynamic model.

2) From Fig.2 the model river line seems very different from the natural water course. The authors should clearly describe how it was derived.

3) About the hydrodynamic model, more details and clarifications are necessary.

3.a) First, the authors state that Bahadurabad is along the Brahmaputra river, but in Figure 1 it seems outside the contour of the basin. If we suppose that the gauged site is available inside the basin near the outlet (and hence, the contour is wrong), it could be sufficient for calibrating the rainfall-runoff model. Why do the authors extent the rainfall-runoff model to the Gange Basin? Moreover, how do they transfer the parameters for the 11 subcatchments to the remaining ones? Please specify.

3.b) About the hydrodynamic model, the procedure of calibration of the cross section geometry is not clear. If Cryosat-2 and Envisat do not refer to the same cross-section (VS), it should be specified how step 1 and step 2 should be applied. Indeed, some details are given in Table 1, but I believe that a deeper description should be added in the text.

Moreover, after the second calibration step, in Fig.3 the flow chart indicates that the procedure is iterative. I do not understand at what level the iteration happens. I think that in order to obtain a calibration the objective function should be unique and minimize

the RMSE for both the steps in parallel. I think this is a very important part of the procedure, therefore I suggest to add details and clarifications. Indeed, page 10 Lines 28-30 should be moved in this section.

3.c) In the hydraulic model, no mention is given to the roughness manning coefficient. Even if it was not specified in the text, I think the authors used a unique coefficient value for the entire river. Please add some details.

3.d) How do you set the initial condition of the model? What about the boundary condition at the downstream site? Please specify.

4) Which is the length of the river simulated with the hydraulic model?

Results:

1) Why do you choose 20 m for defining the outliers of the Cryosat-2 values?

2) The authors state that the manning's number is calibrated. Which is the value? Is it plausible for this river?

3) In the text, it is mentioned that the investigated river reach is the Assam Valley. Figure 7 shows the water levels for a river reach from ∼1950 km to 2800 km. Figure 8 shows the VS at 2839.019. Could the authors add the length of the analyzed river (not well specified) and update Figure 7 for the actual length?

Conclusions:

1) The authors state that "SRTM products do not provide sufficient information to create a hydrodynamic model reproducing accurate water levels or inundations areas". I believe the river is not enough gauged to evaluate the performance of SRTM. In a different study area, the authors could evaluate the accuracy of SRTM in comparison with the proposed procedure, but in this case the only conclusion that can be drawn is that SRTM and radar altimetry gave different results.

2) Could the procedure be transferable to other case studies? Could the authors suggest the minimum width to apply it?

TECHNICAL CORRECTIONS:

Please, remove capital letter after the colon.

Page 3, Line 19: "Mike 11 software": a previous citation of the hydraulic model MIKE 11 used for the analysis is necessary. Please specify if it is a hydrological or hydraulic model and add some references.

Table 1: why 27 cross sections? The Envisat tracks are 13 as reported in the pages 8 Line 15.

References

1) Domeneghetti A., Tarpanelli A., Brocca L., Barbetta S., Moramarco T., Castellarin A., Brath A. (2014) The use of remote sensing-derived water surface data for hydraulic model calibration. Remote Sensing of Environment, 149, 130-141. http://dx.doi.org/10.1016/j.rse.2014.04.007

2) Domeneghetti A., Castellarin A.,Tarpanelli A., Moramarco T. (2015) Investigating the uncertainty of satellite altimetry products for hydrodynamic modelling. Hydrological Processes, 29(23), 4908-4918. http://dx.doi.org/10.1002/hyp.10507

3) Siddique-E-Akbor, A. H., Hossain, F., Lee, H., & Shum, C. K. (2011). Intercomparison study of water level estimates derived from hydrodynamic–hydrologic model and satellite altimetry for a complex deltaic environment. Remote Sensing of Environment, 115, 1522–1531.

4) Tourian M.J., Tarpanelli A., Elmi O., Qin T., Brocca L., Moramarco T., Sneeuw N. (2016) Spatiotemporal densification of river water level time series by multimission satellite altimetry. Water Resources Research, 52. http://dx.doi.org/10.1002/2015WR017654

5) Wilson, M.D., Bates, P. D., Alsdorf, D., Forsberg, B., Horritt, M., Melack, J., et al.

(2007). Modeling large-scale inundation of Amazonian seasonally flooded wetlands. Geophysical Research Letters, 34,L15404. http://dx.doi.org/10.1029/2007GL030156.

6) Yan K., Tarpanelli A., Balint G., Moramarco T., Di Baldassarre G. (2014) Exploring the potential of radar altimetry and SRTM Topography to Support Flood Propagation Modeling: the Danube Case Study. Journal of Hydrologic Engineering 20(2). http://dx.doi.org/10.1061/(ASCE)HE.1943-5584.0001018
* * *

---

## Author Comment (AC1) · 19 Sep 2016

Response to the review of Anonymous Referee #1.

GENERAL COMMENTS: The paper is interesting because it shows a practical use of Cryosat-2 data for a hydrodynamic modelling. So far, a few studies are available on this issue in the scientific literature. Therefore, I found the paper highly timely and appealing. The manuscript is well written and easy to follow, even if some aspects should be better clarified. The main issues concern: 1) the specification of the paper purpose, 2) the description of the hydrodynamic model and 3) the procedure of optimization of the cross-section geometry. Moreover, I have doubts concerning the study area characteristics. The evaluation of the Cryosat-2 data performances cannot be exhaustively tackled if no data are available for the validation.

[Figure]

Reply: We thank the reviewer for constructive feedback on this article. Below, we explain in detail how the individual issues will be solved in the revision.

SPECIFIC COMMENTS:

Introduction: 1) The purpose of the study is not well specified. I suggest the authors to add in the introduction a couple of sentences on this aspect also to introduce the model and the datasets used: why do they use 1D model for this complex river? Why software MIKE 11? Why Cryosat-2 and Envisat?

Reply: The choice of CryoSat-2 is due to its unique drifting orbit which provides water level profiles with high spatial resolution. In combination with (any, not necessarily Envisat) repeat orbit altimetry data providing water level time series at virtual stations these data can be used to calibrate the water level dynamics (which hopefully also becomes more clear in section 3.4). So, the purpose of the paper is to find out what CryoSat-2 can do for river modelling. The 1D model was used because, for the study area, we lack (access to) precise DEMs or bathymetry data. Hence, a 1D model with synthetic cross sections was used – the focus of the study is to simulate water levels (and discharge) in the river, however not flood extent. Furthermore, at the large scale of the model (1000 km plus of river) anything else than a 1D model will become computationally heavy in calibration (and later data assimilation experiments) Plan for revision: Extend the first paragraph of section 1 to include more details on the purpose.

2) I believe that the background should be addressed following the purpose of the paper. The literature review described in the introduction is quite extensive, but it should be more focused on the use of radar altimetry for the calibration of the hydrodynamic models or the cross-sections geometry, mentioning similar studies (see references).

Reply: Thanks for the list of interesting and relevant references. Plan for revision: We will incorporate those into the literature review part.

For example: Domeneghetti et al. (2014; 2015) compared the performances and analyzed the uncertainty of ERS-2 and ENVISAT radar altimetry in the calibration of the manning coefficient of the Hec-RAS model along a river reach of the Po river in Italy. Yan et al. (2014) calibrated the manning roughness coefficient and the depth of the cross sections for the LISFLOOD-FP model in the Danube River with the use of water surface level derived by Envisat radar altimetry. Biancamaria et al. (2009) compared the water levels derived by 22 TOPEX/POSEIDON VSs with the ones simulated by large scales coupled hydrological-hydraulic model of the Ob river in Siberia calibrating the river depth and Manning' roughness coefficient.

3) I suggest citing Tourian et al. (2016) for the merging of satellite altimetry. They analyzed different time series from Envisat, Saral/Altika, Topex/Poseidon and Cryosat-2 in the Po, Congo, Mississippi and Danube rivers.

Study area: 1) Why do the authors focus on Brahmaputra River? Cryosat-2 data are available for rivers where the in-situ data could be easily obtained. The risk to use a poorly gauged river (or as in this case a river where the data are not publicly available) is to be not able to validate the procedure in a proper manner.

Reply: Yes, that is correct, the data over the Brahmaputra River will be hard to validate. The alternative would have been to use another, better gauged river. The choice of suitable rivers however is not very large, because it needs to be of sufficient width, preferably flowing in west-east direction and (fairly) unregulated. One common example is the Amazon River. In this case however the river is being monitored on the ground, hence the information gained from satellite altimetry is less crucial. In this tradeoff it was decided to go for a study region where in-situ data actually is lacking, making application of remote sensing data crucial. Also, for the Amazon River, there are also plenty of altimetry studies available.

2) I have doubts on the use of "calibration" term in the text: "discharge calibration" or "water level calibration". The calibration is referred to the parameters of the model in order to reproduce the measured discharge or water level. I guess that, in this case,

the authors calibrate the parameters of the hydrodynamic model and, then, compare the simulated discharge with the observed one. Therefore, I suggest to pay attention.

Reply: The authors are not sure they entirely understand the comment. Maybe the reviewer could try to clarify?

Data and Methods: 1) This section is quite unbalanced. The description of the satellite data, especially for the water mask, is too long with respect to the hydrodynamic model.

Reply: The authors consider the review of the different filtering methods/water masks used by the inland altimetry databases (paragraph 2 of section 3.1.1) interesting and relevant. Does the reviewer disagree? If so, why? Paragraphs 3 and 4 of section 3.1.1 however will be shortened. Plan for revision: Shorten paragraph 3 and 4 of section 3.1.1 (and expand the hydrodynamic model description, also see below)

2) From Fig.2 the model river line seems very different from the natural water course. The authors should clearly describe how it was derived.

Reply: This disagreement between river mask (∼natural water course) and 1D model river line comes from the fact that the river line is derived from the SRTM DEM by hydrologic routing performed in ArcGIS. Such a course will deviate from the natural water course for Brahmaputra River in the Assam valley mainly because of i) inaccuracies in the SRTM and the relatively flat river valley and ii) changes in the river's course over the years since the acquisition of the SRTM data in 2000. (The hydrodynamic model however will be insensitive to changes in the river's course, as long as the total length of the stretch remains approximately the same) Plan for revision: State the disagreement (and reasons for it) more clearly in section 3.3 where it is mentioned that the 1D model river line is derived from the SRTM DEM.

3) About the hydrodynamic model, more details and clarifications are necessary. 3.a) First, the authors state that Bahadurabad is along the Brahmaputra river, but in Figure 1 it seems outside the contour of the basin. If we suppose that the gauged site is

available inside the basin near the outlet (and hence, the contour is wrong), it could be sufficient for calibrating the rainfall-runoff model. Why do the authors extent the rainfallrunoff model to the Gange Basin? Moreover, how do they transfer the parameters for the 11 subcatchments to the remaining ones? Please specify.

Reply: Bahadurabad station is assumed to be placed at the outlet of the model. It is correct that the outline of the model displayed in Figure 1, which is the basis for the rainfall-runoff model subcatchments, does not agree when the location of the river line is considered in this part of the model. The reason for this is inaccuracies in the SRTM DEM that was used for the subcatchment delineation. Near Bahadurabad station, where the river valley is very flat this lead to the subcatchments ending approximately 10 to 20 km west of where they actually would be expected to end – when one looks at the actual location of the river. However, in the model (remember also that the NAM rainfall-runoff model is a lumped model), the subcatchments are correctly attributed to/draining into the river of the hydrodynamic model. The Brahmaputra basin model used here is, as mentioned in the beginning of section 3.3.1, part of a larger model covering both the Ganges and Brahmaputra basins. This model originates from the Danish hydrologic Institute (DHI). However, no part of the Ganges basin model is used for the work described in the article. So, the model was not really extended to the Ganges basin as the reviewer understood, but it was more that the available information from the Ganges basin (i.e. the rainfall-runoff model parameters) were used for the Brahmaputra basin. Parameters were transferred between subcatchments using simple heuristic rules. Because of the unfortunate situation that only 11 (out of 86 in total) subcatchments could be calibrated against in-situ discharge at their outlets, the other subcatchments had to be given parameters that were derived from those 11. Parameters in the NAM model have some physical meaning, so differences in topography for example can guide in how to transfer parameters from one catchment to another. Furthermore, total runoff from all the aggregated Brahmaputra catchments could be checked against the discharge at Bahadurabad station – the total water balance bias between simulated and observed discharge at that station is only 2%, as mentioned at

the end of section 4.2. Plan for review: Try to make the difference between the Ganges- and Brahmaputra basin more clear and add a few explaining words on the transfer of NAM model parameters.

3.b) About the hydrodynamic model, the procedure of calibration of the cross section geometry is not clear. If Cryosat-2 and Envisat do not refer to the same cross-section (VS), it should be specified how step 1 and step 2 should be applied. Indeed, some details are given in Table 1, but I believe that a deeper description should be added in the text.

Reply: Yes, the different number of cross sections is confusing. The data from Envisat does not cover exactly the same river section as the CryoSat-2 data used, hence not all the same cross sections are calibrated. With Envisat, cross sections from river km 2050 to 3050 could be calibrated, whilst with CryoSat-2 data cross section between rive km 1950 and 2800. Hence, only the overlap from river km 2050 to 2800 can be considered fully calibrated. Plan for revision: state the sections of the river with overlapping satellite data from Envisat and CryoSat-2 clearly (also rework Table 1). Furthermore, Figure 3 (the flowchart of the whole calibration process) will be reworked for the revision, hopefully adding more clarity.

Moreover, after the second calibration step, in Fig.3 the flow chart indicates that the procedure is iterative. I do not understand at what level the iteration happens. I think that in order to obtain a calibration the objective function should be unique and minimize the RMSE for both the steps in parallel. I think this is a very important part of the procedure, therefore I suggest to add details and clarifications. Indeed, page 10 Lines 28-30 should be moved in this section.

Reply: Yes, the authors agree that Figure 3 can be improved. We however are not sure how to go about minimizing the RMSE for both steps in parallel, as suggested by the reviewer. The two different objectives from step 1 and step 2 could be merged into one optimization. This however would also increase the complexity of the problem, as both

sets of decision variables would have to be considered. This probably will increase the computational demand of the already demanding optimization problem. Hence, because the especially the sensitivity of the water level amplitudes to small changes in the cross section datums is very low, this iterative approach is considered sufficient. Given the low sensitivity of, in other words, the objective of step 2 to changes resulting from step 1, the iteration usually can be ended after a run of step 1. Plan for revision: Explain the iterative calibration process in more detail as outlined in the reply above, and improve Figure 3 for more clarity.

3.c) In the hydraulic model, no mention is given to the roughness manning coefficient. Even if it was not specified in the text, I think the authors used a unique coefficient value for the entire river. Please add some details.

Reply: The Manning coefficient was calibrated to one unique value along the entire river. Plan for revision: Include a few lines on the manning coefficient calibration and the resulting value.

3.d) How do you set the initial condition of the model? What about the boundary condition at the downstream site? Please specify.

Reply: The initial conditions of the hydrodynamic model are taken from a hotstart of the model. More important however is the hotstart of the hydrologic part of the mode, i.e. the NAM rainfall-runoff models, because those models have states with much "longer memory". The hotstart of the NAM rainfall-runoff models was created by running the initial calibration period of the model, 2002 to 2007, 30 times, reusing the final state of the respective prior run as a hotstart. After 30 iterations, or 180 years, all states with long memory (mainly groundwater and snow storage) have reached steady-state. This steady state then is used as the hotstart. A time series of water levels at the downstream boundary of the model (which lies ~180 km downstream of Bahadurabad station) could be obtained for the years 2001 to 2009 – outside that range a climatology of these values was obtained. In any case, discharge and water level at Bahadurabad

station (so the downstream end of the area of interest) is insensitive to this downstream boundary condition. Plan for revision: Include these details in the text.

4) Which is the length of the river simulated with the hydraulic model?

Plan for revision: Include details in the text.

Results: 1) Why do you choose 20 m for defining the outliers of the Cryosat-2 values?

Reply: This is a value that was chosen after inspecting the differences between SRTM and CryoSat-2 elevations along the river. In general, CryoSat-2 observations group nicely around the SRTM values, and only very few clear outliers do exist – these will be removed with the chosen threshold of 20 metres deviation.

2) The authors state that the manning's number is calibrated. Which is the value? Is it plausible for this river?

Reply: The resulting value is a Manning's n value of 0.029, which is slightly low for the given river, but should be considered plausible (compare for example http://www.fsl.orst.edu/geowater/FX3/help/8_Hydraulic_Reference/Mannings_n_Tables.htm) Plan for revision: (also see above) Include a few words on the manning coefficient calibration.

3) In the text, it is mentioned that the investigated river reach is the Assam Valley. Figure 7 shows the water levels for a river reach from 1950 km to 2800 km. Figure 8 shows the VS at 2839.019. Could the authors add the length of the analyzed river (not well specified) and update Figure 7 for the actual length?

Plan for revision: This will be updated in accordance to what is written as reply to comment 3.b) above.

Conclusions: 1) The authors state that "SRTM products do not provide sufficient information to create a hydrodynamic model reproducing accurate water levels or inundations areas". I believe the river is not enough gauged to evaluate the performance of

SRTM. In a different study area, the authors could evaluate the accuracy of SRTM in comparison with the proposed procedure, but in this case the only conclusion that can be drawn is that SRTM and radar altimetry gave different results.

Reply: The authors agree that this statement may be too simplistic. However, the authors still believe that the SRTM – at least as a raw product – is not precise enough to directly derive a hydrodynamic model accurately reproducing water levels. This can be shown for example by the significant improvements that Jarihani et al. (2015) could achieve when deriving cross sections from SRTM DEM, and then subsequently correcting the SRTM DEM for vegetation and other issues (Table 4 in their article). Their baseline for the comparison is derived from ICESat data, which they could validate against in-situ data to have a RMSD of only 0.23m. But even when the SRTM DEM was vegetation smoothed and hydrologically corrected, its RMSD compared to a cross section from ICESat was above 1.1m. Another example is the work by Md Ali et al. (2015) using a DEM from lidar data with 1m resolution to set up a 1D hydrodynamic model and comparing it to, amongst others, the same hydrodynamic model based on the SRTM DEM. They found the resulting simulated water levels of the SRTM DEM based hydrodynamic model to have a MAD of 0.76m compared to the same levels from the lidar based model. With the proposed procedure, the water levels will be fitted to CryoSat-2 observations. Based on the literature cited the authors assume that fitting the simulated water levels to CryoSat-2 data also means a better fit to real water levels than what can be achieved by setting up the hydrodynamic model based on the SRTM DEM only. Remember also the difficulties of obtaining an estimate of bathymetry from DEMs, whilst the suggested procedure does not require any knowledge of bathymetry. References: Jarihani, A. A., Callow, J. N., McVicar, T. R., Van Niel, T. G. and Larsen, J. R.: Satellite-derived Digital Elevation Model (DEM) selection, preparation and correction for hydrodynamic modelling in large, low-gradient and data-sparse catchments, J. Hydrol., 524, 489–506, doi:10.1016/j.jhydrol.2015.02.049, 2015. Md Ali, A., Solomatine, D. P. and Di Baldassarre, G.: Assessing the impact of different sources of topographic data on 1-D hydraulic modelling of floods, Hydrol. Earth Syst. Sci., 19,

631–643, doi:10.5194/hess-19-631-2015, 2015. Plan for revision: Reformulate that statement, stating the assumptions made, including the references mentioned.

2) Could the procedure be transferable to other case studies? Could the authors suggest the minimum width to apply it?

Reply: Yes, the authors expect that this procedure can be transferred to other case studies.. A minimum river width however seems to be hard to define, as the ability of satellite altimeters to reliably measure water level in (narrow) rivers depends (besides the actual instrument and processing) not only on the river width, but also on the topography of the river valley – see for example what is discussed in connection with the results shown in Figure 5 and the article by Dehecq et al., 2013. Plan for revision: Add a few words on the transferability of the developed procedure.

TECHNICAL CORRECTIONS: Please, remove capital letter after the colon. Page 3, Line 19: "Mike 11 software": a previous citation of the hydraulic model MIKE 11 used for the analysis is necessary. Please specify if it is a hydrological or hydraulic model and add some references.

Plan for revision: Yes, this will be included in the revision.

Table 1: why 27 cross sections? The Envisat tracks are 13 as reported in the pages 8 Line 15.

Reply: Yes, there exist only 13 virtual stations along the Assam valley. Those virtual stations however can be used to calibrate both neighbouring cross sections (as the virtual station usually is not placed directly on a cross section). Furthermore, angles for cross sections lacking neighbouring virtual stations were linearly interpolated between the next cross sections. Plan for revision: Clarify these things together with what is mentioned in the reply to comment 3.b)

References 1) Domeneghetti A., Tarpanelli A., Brocca L., Barbetta S., Moramarco T., Castellarin A., Brath A. (2014) The use of remote sensing-derived water surface data for hydraulic model calibration. Remote Sensing of Environment, 149, 130-141. http://dx.doi.org/10.1016/j.rse.2014.04.007 2) Domeneghetti A., Castellarin A.,Tarpanelli A., Moramarco T. (2015) Investigating the uncertainty of satellite altimetry products for hydrodynamic modelling. Hydrological Processes, 29(23), 4908-4918. http://dx.doi.org/10.1002/hyp.10507 3) Siddique-E-Akbor, A. H., Hossain, F., Lee, H., & Shum, C. K. (2011). Intercomparison study of water level estimates derived from hydrodynamic–hydrologic model and satellite altimetry for a complex deltaic environment. Remote Sensing of Environment, 115, 1522–1531. 4) Tourian M.J., Tarpanelli A., Elmi O., Qin T., Brocca L., Moramarco T., Sneeuw N. (2016) Spatiotemporal densification of river water level time series by multimission satellite altimetry. Water Resources Research, 52. http://dx.doi.org/10.1002/2015WR017654 5) Wilson, M.D., Bates, P. D., Alsdorf, D., Forsberg, B., Horritt, M., Melack, J., et al. (2007). Modeling large-scale inundation of Amazonian seasonally ïnËĞC' ooded wetlands. Geophysical Research Letters, 34,L15404. http://dx.doi.org/10.1029/2007GL030156. 6) Yan K., Tarpanelli A., Balint G., Moramarco T., Di Baldassarre G. (2014) Exploring the potential of radar altimetry and SRTM Topography to Support Flood Propagation Modeling: the Danube Case Study. Journal of Hydrologic Engineering 20(2). http://dx.doi.org/10.1061/(ASCE)HE.1943-5584.0001018

---

## Referee Comment (RC2) · Anonymous Referee #2 · 20 Sep 2016

This paper describes the application of remotely sensed altimetry data from the CryoSat-2 satellite to large scale hydraulic modelling, using the Brahmaputra Basin as an example. While the paper is generally well written and clear, there are a few issues related to the focus and balance of the paper that will need addressing.

The remote sensing aspects of the study seem very well described, but the description of the hydraulic modelling is relatively weak. In this respect, the novelty of the work lies in the use of the Cryosat-2 data rather than the hydraulic modelling. In fact given the current research in large scale hydraulic modelling the approach used in the paper is overly simple. Moving beyond the "virtual gauge" is of great research interest and I think this study has real value here, particularly with the fusion of drifting orbit and Envisat virtual stations. The filtering using a dynamic Landsat water mask is also of value and overall I think there is sufficient novelty in the work for publication.

[Figure]

While there are some issues to address, I do not think further modelling is required. I think most of the issues can be addressed with changes to the core text. There should be better reference to existing large scale hydraulic river modelling and more discussion/openness about the modelling limitations.

Some more specific points that should be addressed:

(1) The work seems to miss some aspects of recent research that I would assume would be relevant to the work. For example no mention is made of studies that use ICESAT – another dataset that has been used for similar hydraulic model calibration. There is also no reference to the relevant work on channel representation in large scale 1d-2d modelling such as that of Neal et al (2015) (and previous studies).

(2) Why only use a 1d model when there are plenty examples of this scale of hydraulic model using 1d&2d? Essentially all the floodplain and braided river section details are being lumped into the single triangular cross-section, so I am not sure how valid the representation of the river/floodplain is in the end. It might work as a simple water level response function that can be calibrated (as demonstrated in the paper), but it losses any physically based reality in representing the river and its floodplain, thereby limiting the value to the model for basin/river/floodplain studies. It is possible of course that the hydraulic conditions are such that the detailed representation of the channel is less important, such as found by Trigg et al 2009 on the Amazon. However there is no detail provided to show this is the case, for example what are the Froude numbers for the flow? It has not been demonstrated that the resulting model has value outside of the modelled scenario. I don't think that the model necessarily has to be redone, but I do think its limitations need more discussion.

(3) More discussion is required on the uncertainty in flow produced by the rainfall runoff modelling and how it affects the hydraulic modelling.

(4) There is reference to the dynamic nature of river channel with regards to the water mask, but no discussion of the how important this geomorphology might be to the

simple triangle river channel model used.

(5) I am not clear on how the SRTM is actually translated into the triangle river channel. Has the raw SRTM data been processed to remove the vegetation bias? What is actually used for the 1d triangle, the width and depth of the river extracted from the SRTM? If so maybe river width from landsat would be better for the width and estimate of depth from geomorphological relationships (Leopold, and Maddock, 1953) would be better? What size are these calibrated triangles. Do they bear any resemblance to the real river sections?

(6) Manning's is mentioned but no values given. Given its direct control on water levels and it should have some link to expected values it should not be omitted. Given the crude nature of the cross-sections and the fact that Manning's will compensate for lots of missing processes in this regard, I am not sure the calibrated Manning's values will bear resemblance to what might be expected for such a river.

Refs: Neal et al, 2015. Efficient incorporation of channel cross-section geometry uncertainty into regional and global scale flood inundation models, Journal of Hydrology. Trigg et al 2009. Amazon flood wave hydraulics. Journal of Hydrology. Leopold, and Maddock, 1953, The hydraulic geometry of stream channels and some physiographic implications, U.S. Geol. Surv. Prof. Pap.

---

## Author Comment (AC2) · 24 Sep 2016

Reply to the review of Anonymous Referee #2

This paper describes the application of remotely sensed altimetry data from the CryoSat-2 satellite to large scale hydraulic modelling, using the Brahmaputra Basin as an example. While the paper is generally well written and clear, there are a few issues related to the focus and balance of the paper that will need addressing.

The remote sensing aspects of the study seem very well described, but the description of the hydraulic modelling is relatively weak. In this respect, the novelty of the work lies in the use of the Cryosat-2 data rather than the hydraulic modelling. In fact given the current research in large scale hydraulic modelling the approach used in the paper is overly simple. Moving beyond the "virtual gauge" is of great research interest and

[Figure]

I think this study has real value here, particularly with the fusion of drifting orbit and Envisat virtual stations. The filtering using a dynamic Landsat water mask is also of value and overall I think there is sufficient novelty in the work for publication.

While there are some issues to address, I do not think further modelling is required. I think most of the issues can be addressed with changes to the core text. There should be better reference to existing large scale hydraulic river modelling and more discussion/openness about the modelling limitations.

Reply: We thank the reviewer for constructive and insightful comments and suggestions. We fully agree with the reviewerthat the contribution of this paper is the integration of CryoSat-2 data into a hydrodynamic model (and not for the hydrologic-hydrodynamic modelling as such).

Some more specific points that should be addressed: (1) The work seems to miss some aspects of recent research that I would assume would be relevant to the work. For example no mention is made of studies that use ICESAT – another dataset that has been used for similar hydraulic model calibration. There is also no reference to the relevant work on channel representation in large scale 1d-2d modelling such as that of Neal et al (2015) (and previous studies).

Reply: We agree that the referencing may be too narrow in places and will include the suggested context in the revision.

Plan for revision: Include and discuss further references, as pointed out by reviewer.

(2) Why only use a 1d model when there are plenty examples of this scale of hydraulic model using 1d&2d? Essentially all the floodplain and braided river section details are being lumped into the single triangular cross-section, so I am not sure how valid the representation of the river/floodplain is in the end. It might work as a simple water level response function that can be calibrated (as demonstrated in the paper), but it losses any physically based reality in representing the river and its floodplain, thereby limiting

the value to the model for basin/river/floodplain studies. It is possible of course that the hydraulic conditions are such that the detailed representation of the channel is less important, such as found by Trigg et al 2009 on the Amazon. However there is no detail provided to show this is the case, for example what are the Froude numbers for the flow? It has not been demonstrated that the resulting model has value outside of the modelled scenario. I don't think that the model necessarily has to be redone, but I do think its limitations need more discussion.

Reply: Focus here is on accurate prediction of water levels and discharge, this is not a flood model. The main reason for choosing a 1D model was computational efficiency. It is correct that 1d-2d modelling at this scale is feasible, but probabilistic approaches using large ensembles of model runs would pose significant computational challenges. For example, the cross section calibration presented in the article using a genetic algorithm to find optimal parameters requires many runs (in the range of 10 000) of the model. Moreover, running a meaningful 1d-2d model would require accurate topography/bathymetry, which is unavailable for this braided and highly dynamic river system. The result of the cross section calibration, especially of step 2 where the amplitudes are being fitted, is consistent with the results by Trigg et al. (2009): with the chosen – simplistic – cross section representation we are able to reproduce observed water level dynamics. The authors however assume that the study is transferable to other rivers as well, given the availability of sufficient altimetry data. Even if a triangular cross section with varying angle will not be able to reproduce observed water level dynamics for all rivers, the same calibration procedure could be applied to other descriptions of cross section geometry. For example the same procedure should also work for the power-function cross section shape described by Neal et al. (2015): Instead of using a triangular cross section with the angle as only parameter, one could use the power function cross section shape and use i) only shape parameter s or also ii) both shape parameter s and bankfull depth h_full as calibration parameters. (assuming that the third parameter to describe the cross section in Neal et al.'s approach, bankfull width w_full, can easily be estimated from remote sensing data)

Plan for revision: Include some discussion as initiated by the reviewer and include some details on the hydraulic model and results as well as limitations

(3) More discussion is required on the uncertainty in flow produced by the rainfall runoff modelling and how it affects the hydraulic modelling.

Plan for revision: Include some details and discussion of the uncertainty of the rainfall-runoff models. Potentially including data from the calibration catchments.

(4) There is reference to the dynamic nature of river channel with regards to the water mask, but no discussion of the how important this geomorphology might be to the simple triangle river channel model used.

Reply: For this simple 1D model, with its synthetic cross sections, we assume that the change of the river channels does not significantly affect the water level-discharge relationships. Also, Mirza (2003) found rating curves at the Brahmaputra to exhibit fairly constant Q-h relationships over decades, leading to the conclusion that the Brahmaputra River is in "dynamic equilibrium".

Plan for revision: Include some discussion of this

Reference: Mirza, M. M. Q.: The Choice of Stage-Discharge Relationship for the Ganges and Brahmaputra Rivers in Bangladesh, Nord. Hydrol., 34(4), 321–342, doi:10.2166/nh.2003.019, 2003.

(5) I am not clear on how the SRTM is actually translated into the triangle river channel. Has the raw SRTM data been processed to remove the vegetation bias? What is actually used for the 1d triangle, the width and depth of the river extracted from the SRTM? If so maybe river width from landsat would be better for the width and estimate of depth from geomorphological relationships (Leopold, and Maddock, 1953) would be better? What size are these calibrated triangles. Do they bear any resemblance to the real river sections?

Reply: What is referred to as "reference" cross sections in the paper was visually extracted from satellite imagery and the SRTM DEM in a consulting project preparing the Ganges-Brahmaputra hydrologic model used in this paper. The real river cross sections (of this multi-channelled river) will of course be very different from these simplistic cross sections. However, this is not so important in a 1D model, as long as the relationship $A = A(h)$ and $P = P(h)$ are realistic, i.e. we need to get the relationship between flow cross sectional area and wetted parameter right.

Plan for revision: With this context, maybe the term "reference" cross sections is a bit bold; and this limitation should be discussed when showing the "reference" water levels in Figure 7.

(6) Manning's is mentioned but no values given. Given its direct control on water levels and it should have some link to expected values it should not be omitted. Given the crude nature of the cross-sections and the fact that Manning's will compensate for lots of missing processes in this regard, I am not sure the calibrated Manning's values will bear resemblance to what might be expected for such a river.

Reply: Manning's number only has been calibrated against discharge (timing of discharge) at the outlet station Bahadurabad. Its value is 0.029, which is slightly low but can be considered plausible – see for example http://www.fsl.orst.edu/geowater/FX3/help/8_Hydraulic_Reference/Mannings_n_Tables.htm.

Plan for revision: Include a short description of the Manning calibration; and also extend Figure 3 of the article to make the different calibration steps more clear (this is also something pointed out by another reviewer)

Refs: Neal et al, 2015. Efficient incorporation of channel cross-section geometry uncertainty into regional and global scale flood inundation models, Journal of Hydrology. Trigg et al 2009. Amazon flood wave hydraulics. Journal of Hydrology. Leopold, and Maddock, 1953, The hydraulic geometry of stream channels and some physiographic implications, U.S. Geol. Surv. Prof. Pap.

---

## Author Response (AR1)

Review by anonymous referee #1

*GENERAL COMMENTS:*

| Reviewer's comment | *The paper is interesting because it shows a practical use of Cryosat-2 data for a hydrodynamic modelling. So far, a few studies are available on this issue in the scientific literature. Therefore, I found the paper highly timely and appealing.*
*The manuscript is well written and easy to follow, even if some aspects should be better clarified. The main issues concern: 1) the specification of the paper purpose, 2) the description of the hydrodynamic model and 3) the procedure of optimization of the cross-section geometry. Moreover, I have doubts concerning the study area characteristics. The evaluation of the Cryosat-2 data performances cannot be exhaustively tackled if no data are available for the validation.* |
|---|---|
| Authors' response | We thank the reviewer for constructive feedback on this article. We believe incorporating the reviewer's comments has improved the article. |
| changes | - |

*SPECIFIC COMMENTS:*

*Introduction:*

| Reviewer's comment | *1) The purpose of the study is not well specified. I suggest the authors to add in the introduction a couple of sentences on this aspect also to introduce the model and the datasets used: why do they use 1D model for this complex river? Why software MIKE 11? Why Cryosat-2 and Envisat?* |
|---|---|
| Authors' response | The choice of CryoSat-2 is due to its unique drifting orbit which provides water level profiles with high spatial resolution. In combination with (any, not necessarily Envisat) repeat orbit altimetry data providing water level time series at virtual stations these data can be used to calibrate the water level dynamics (which hopefully also becomes more clear in section 3.4). So, the purpose of the paper is to find out what CryoSat-2 can do for river modelling.
The 1D model was used because, for the study area, we lack (access to) precise DEMs or bathymetry data. Hence, a 1D model with synthetic cross sections was used – the focus of the study is to simulate water levels (and discharge) in the river, however not flood extent. Furthermore, at the large scale of the model (1000 km plus of river) anything else than a 1D model will become computationally heavy in calibration (or potential use of model ensembles in uncertainty analysis or data assimilation). |
| changes | p. 1, line 29 – p. 2, line 3: extended motivation in introduction
p. 3, line 31 – p. 4, line 10: new section 1.3 on Hydrodynamic river models (mainly as response to comment 2 of anonymous referee #2) |

| Reviewer's comment | 2) I believe that the background should be addressed following the purpose of the paper. The literature review described in the introduction is quite extensive, but it should be more focused on the use of radar altimetry for the calibration of the hydrodynamic models or the cross-sections geometry, mentioning similar studies (see references).
For example:
Domeneghetti et al. (2014; 2015) compared the performances and analyzed the uncertainty of ERS-2 and ENVISAT radar altimetry in the calibration of the manning coefficient of the Hec-RAS model along a river reach of the Po river in Italy.
Yan et al. (2014) calibrated the manning roughness coefficient and the depth of the cross sections for the LISFLOOD-FP model in the Danube River with the use of water surface level derived by Envisat radar altimetry.
Biancamaria et al. (2009) compared the water levels derived by 22 TOPEX/POSEIDON VSs with the ones simulated by large scales coupled hydrological-hydraulic model of the Ob river in Siberia calibrating the river depth and Manning' roughness coefficient. |
|---|---|
| Authors' response | Thanks for the list of interesting and relevant references. |
| changes | p. 3, line 6 – line 13: extended literature review |

| Reviewer's comment | 3) I suggest citing Tourian et al. (2016) for the merging of satellite altimetry. They analyzed different time series from Envisat, Saral/Altika, Topex/Poseidon and Cryosat-2 in the Po, Congo, Mississippi and Danube rivers. |
|---|---|
| Authors' response | Indeed, a relevant reference |
| changes | p. 2, line 19 – 23: included the reference and some discussion of it |

*Study area:*

| Reviewer's comment | 1) Why do the authors focus on Brahmaputra River? Cryosat-2 data are available for rivers where the in-situ data could be easily obtained. The risk to use a poorly gauged river (or as in this case a river where the data are not publicly available) is to be not able to validate the procedure in a proper manner. |
|---|---|
| Authors' response | Yes, that is correct, the data over the Brahmaputra River is hard to validate. The alternative would have been to use another, better gauged river. The choice of suitable rivers however is not very large, because it needs to be of sufficient width, preferably flowing in west-east direction and (fairly) unregulated. One common example is the Amazon River. In this case however the river is being monitored on the ground, hence the information gained from satellite altimetry is less crucial. Furthermore, the Amazon River (in large parts) is exceptionally wide, and not representative of other rivers (if one considers the transferability of this work). In this trade-off it was decided to go for a study region where in-situ data actually is scarce, making application of remote sensing data crucial. Also, for the Amazon River, there are already plenty of altimetry studies available. |
| changes | p. 4, line 20 – 23: extended section 2 Study area |

| Reviewer's comment | 2) I have doubts on the use of "calibration" term in the text: "discharge calibration" or "water level calibration". The calibration is referred to the parameters of the model in order to reproduce the measured discharge or water level. I guess that, in this case, the authors calibrate the parameters of the hydrodynamic model and, then, compare the simulated discharge with the observed one. Therefore, I suggest to pay attention. |
|---|---|
| Authors' response | We assume that the confusion is about whether naming the calibration after the target (discharge or water level) or the calibration parameter (for example cross section datum). We hope that the changes described below, together with the extended Figure 3, makes things more clear. |

| changes | p. 1, line 18 and line 17 – 18: clarified the terms in the abstract |
|---|---|
| | p. 4, line 31 – p. 5, line 5: Changed term "water level calibration" to the throughout the manuscript used "cross section calibration", and explained what happened during the (discharge) calibration of the hydrodynamic model |
| | p. 9, line 24 – 27: As above. Also added "fitting water levels to observed water levels from altimetry" to make the calibration target clear. |
| | Furthermore, the captions of Figure 7, Figure 8 and Table 1 were changed ("cross section calibration" instead of "water level calibration") |

*Data and Methods:*

| Reviewer's comment | *1) This section is quite unbalanced. The description of the satellite data, especially for the water mask, is too long with respect to the hydrodynamic model.* |
|---|---|
| Authors' response | This was also pointed out by the other reviewer (and confirmed by the editor): the description of the hydrodynamic model is too short, it was extended. For details see below (comments 3a to 4) |
| changes | See below |

| Reviewer's comment | *2) From Fig.2 the model river line seems very different from the natural water course. The authors should clearly describe how it was derived.* |
|---|---|
| Authors' response | This disagreement between river mask (~natural water course) and 1D model river line comes from the fact that the river line is derived from the SRTM DEM by DEM hydroprocessing performed in ArcGIS. Such a course will deviate from the natural water course for Brahmaputra River in the Assam valley mainly because of i) inaccuracies in the SRTM and the relatively flat river valley and ii) changes in the river's course over the years since the acquisition of the SRTM data in 2000. |
| | (The hydrodynamic model however will be insensitive to changes in the river's course, as long as the total length of the stretch remains approximately the same) |
| | (see also response to comment 3a below and to comment 4 by anonymous referee #2) |
| changes | p. 8, line 5 – 8: added two sentences on this to section 3.3.1 Hydrodynamic model |

| Reviewer's comment | *3) About the hydrodynamic model, more details and clarifications are necessary.* |
|---|---|
| | *3.a) First, the authors state that Bahadurabad is along the Brahmaputra river, but in Figure 1 it seems outside the contour of the basin. If we suppose that the gauged site is available inside the basin near the outlet (and hence, the contour is wrong), it could be sufficient for calibrating the rainfall-runoff model. Why do the authors extent the rainfallrunoff model to the Gange Basin? Moreover, how do they transfer the parameters for the 11 subcatchments to the remaining ones? Please specify.* |

| | |
|---|---|
| Authors' response | Bahadurabad station is assumed to be placed at the outlet of the model. It is correct that the outline of the model displayed in Figure 1, which is the basis for the rainfall-runoff model subcatchments, does not agree when the location of the river line is considered in this part of the model. The reason for this is inaccuracies in the SRTM DEM that was used for the subcatchment delineation. Near Bahadurabad station, where the river valley is very flat this lead to the subcatchments ending approximately 10 to 20 km west of where they actually would be expected to end – when one looks at the actual location of the river. However, in the model (remember also that the NAM rainfall-runoff model is a lumped model), the subcatchments are correctly attributed to/draining into the river of the hydrodynamic model.

The Brahmaputra basin model used here is part of a larger model covering both the Ganges and Brahmaputra basins. This model originates from a consultancy project of DHI and ICIMOD. However, no part of the Ganges basin model is used for the work described in the article. So, the model was not really extended to the Ganges basin as the reviewer might have understood, but rather that the available information from the Ganges basin (i.e. the rainfall-runoff model parameters) was used for the Brahmaputra basin.

Parameters were transferred between subcatchments using simple heuristics. Because of the unfortunate situation that only 11 (out of 86 in total) subcatchments could be calibrated against in-situ discharge at their outlets, the other subcatchments had to be given parameters that were derived from those 11. Parameters in the NAM model have some physical meaning, so differences in topography for example can guide in how to transfer parameters from one catchment to another. Furthermore, total runoff from all the aggregated Brahmaputra catchments could be checked against the discharge at Bahadurabad station – the total water balance bias between simulated and observed discharge at that station is only 2%, as mentioned at the end of section 4.2. |
| changes | As part of the expansion of the model explanation, further subsections were added (now: 3.3.1 Hydrodynamic model, 3.3.2 Rainfall-runoff forcing of the hydrodynamic model, 3.3.3 Boundary and initial conditions) and the heading of section 3.3 was changed to "Hydrologic-hydrodynamic model"
p. 8, line 5 – 9: added some explanations on the inaccuracies of the SRTM DEM causing slight disagreements between river's course, gauging station location, and basin outline
p. 8, line 16 – 17: added some words on the origin/initial setup of the Brahmaputra model
p. 8, line 30 – 33: added some words explaining the NAM parameter transfer
p. 9, line 1 – 2: removed reference to Hardinge Bridge station for discharge calibration, as it is part of the Ganges River which is not part of this study. This might also have caused some confusion to the reviewer.
p. 12, line 30 – 32: added some words on the (good) calibration results and how they indicate that the NAM parameter transfer can be considered successful |

| | |
|---|---|
| Reviewer's comment | *3.b) About the hydrodynamic model, the procedure of calibration of the cross section geometry is not clear. If Cryosat-2 and Envisat do not refer to the same cross-section (VS), it should be specified how step 1 and step 2 should be applied. Indeed, some details are given in Table 1, but I believe that a deeper description should be added in the text.* |
| Authors' response | Yes, the different number of cross sections is confusing. The data from Envisat does not cover exactly the same river section as the CryoSat-2 data used, hence not all the same cross sections are calibrated. With Envisat, cross sections from river km 2050 to 3050 could be calibrated, whilst with CryoSat-2 data cross section between rive km 1950 and 2800. Hence, only the overlap from river km 2050 to 2800 can be considered fully calibrated. In other words, the 22 cross sections in this overlap are calibrated in both step 1 and step 2. |
| changes | Table 1: added river km to the calibration parameters
p. 13, line 18 – 21: added explanation on the overlap
(furthermore, we hope that the changed Figure 3 helps explaining the calibration process better – see the comment below) |

| | |
|---|---|
| *Reviewer's comment* | *Moreover, after the second calibration step, in Fig.3 the flow chart indicates that the procedure is iterative. I do not understand at what level the iteration happens. I think that in order to obtain a calibration the objective function should be unique and minimize the RMSE for both the steps in parallel. I think this is a very important part of the procedure, therefore I suggest to add details and clarifications. Indeed, page 10 Lines 28-30 should be moved in this section.* |
| Authors' response | Yes, the authors agree that Figure 3 can be improved. We however are not sure how to go about minimizing the RMSE for both steps in parallel, as suggested by the reviewer. The two different objectives from step 1 and step 2 could be merged into one optimization. This however would also increase the complexity of the problem, as both sets of decision variables would have to be considered. This probably will increase the computational demand of the already demanding optimization problem. Hence, because the sensitivity of the water level amplitudes to small changes in the cross section datums is very low, this iterative approach is considered sufficient. Given the low sensitivity of, in other words, the objective of step 2 to changes resulting from step 1, the iteration usually can be ended after a run of step 1. |
| changes | p. 13, line 16 – 18 (which was p. 10, line 28 – 30 in the original document) moved and slightly extended as suggested by reviewer to p. 10, line 25 – 31
Figure 3: changed. Extended, to add more clarity (also concerning the discharge and runoff calibration) |

| | |
|---|---|
| *Reviewer's comment* | *3.c) In the hydraulic model, no mention is given to the roughness manning coefficient. Even if it was not specified in the text, I think the authors used a unique coefficient value for the entire river. Please add some details.* |
| Authors' response | The Manning coefficient was calibrated to one unique value along the entire river. |
| changes | p. 8, line 11 – 13: added explanation on Manning's number calibration to section 3.3.1 Hydrodynamic model
(for results, see response to comment 2 in the Results section) |

| | |
|---|---|
| *Reviewer's comment* | *3.d) How do you set the initial condition of the model? What about the boundary condition at the downstream site? Please specify.* |
| Authors' response | The initial conditions of the hydrodynamic model are taken from a hotstart of the model. More important however is the hotstart of the hydrologic part of the mode, i.e. the NAM rainfall-runoff models, because those models have states with much "longer memory". The hotstart of the NAM rainfall-runoff models was created by running the initial calibration period of the model, 2002 to 2007, 30 times, reusing the final state of the respective prior run as a hotstart. After 30 iterations, or 180 years, all states with long memory (mainly groundwater and snow storage) have reached equilibrium. This then is used as the hotstart.
A time series of water levels at the downstream boundary of the model (which lies ~180 km downstream of Bahadurabad station) could be obtained for the years 2001 to 2009 – outside that range a climatology of these values was used. In any case, discharge and water level at Bahadurabad station (so the downstream end of the area of interest) is insensitive to this downstream boundary condition. |
| changes | p. 9, line 3 – 11: added section 3.3.3 Boundary and initial conditions of the model |

| | |
|---|---|
| *Reviewer's comment* | *4) Which is the length of the river simulated with the hydraulic model?* |
| Authors' response | The total length of the Brahmaputra River modelled (tributaries excluded) is approximately 3090 km. |
| changes | p. 7, line 27 – 28: added this to section 3.3.1 Hydrodynamic model |

*Results:*

| | |
|---|---|
| *Reviewer's comment* | *1) Why do you choose 20 m for defining the outliers of the Cryosat-2 values?* |
| Authors' response | This is a value that was chosen after inspecting the differences between SRTM and CryoSat-2 elevations along the river. In general, CryoSat-2 observations group nicely around the SRTM values, and only very few clear outliers -do exist – these will be removed with the chosen threshold of 20 metres deviation. |
| changes | p. 11, line 3 – 8: added explanations on outlier definition. Furthermore, an error with the number of outlier filtered measurements was corrected. |

| | |
|---|---|
| *Reviewer's comment* | *2) The authors state that the manning's number is calibrated. Which is the value? Is it plausible for this river?* |
| Authors' response | The resulting value is a Manning's n value of 0.029 in SI units, which is considered plausible (compare for example http://www.fsl.orst.edu/geowater/FX3/help/8_Hydraulic_Reference/Mannings_n_Tables.htm) (same as comment 6 by anonymous referee #2) |
| changes | p. 12, line 10 – 11: added the above |

| | |
|---|---|
| *Reviewer's comment* | *3) In the text, it is mentioned that the investigated river reach is the Assam Valley. Figure 7 shows the water levels for a river reach from 1950 km to 2800 km. Figure 8 shows the VS at 2839.019. Could the authors add the length of the analyzed river (not well specified) and update Figure 7 for the actual length?* |
| Authors' response | The authors understand that the figures should show the relevant stretch (the overlap between river km 2050 and 2800, see reply to comment 3b).We hope that also the changes made in response to comment 3b) above help clarifying this. |
| changes | Figure 7: changed to show river only between river km 2050 and 2800, i.e. the stretch covered by both Envisat and CryoSat-2 data
Figure 8: changed to VS at river km 2750 |

*Conclusions:*

| | |
|---|---|
| *Reviewer's comment* | *1) The authors state that "SRTM products do not provide sufficient information to create a hydrodynamic model reproducing accurate water levels or inundations areas". I believe the river is not enough gauged to evaluate the performance of SRTM. In a different study area, the authors could evaluate the accuracy of SRTM in comparison with the proposed procedure, but in this case the only conclusion that can be drawn is that SRTM and radar altimetry gave different results.* |

| Authors' response | The authors agree that this statement may be too simplistic. However, the authors still believe that the SRTM – at least as a raw product – is not precise enough to directly derive a hydrodynamic model accurately reproducing water levels. This can be shown for example by the significant improvements that Jarihani et al. (2015) could achieve when deriving cross sections from SRTM DEM, and then subsequently correcting the SRTM DEM for vegetation and other issues (Table 4 in their article). Their baseline for the comparison is derived from ICESat data, which they could validate against in-situ data to have a RMSD of only 0.23m. But even when the SRTM DEM was vegetation smoothed and hydrologically corrected, its RMSD compared to a cross section from ICESat was above 1.1m. Another example is the work by Md Ali et al. (2015) using a DEM from lidar data with 1m resolution to set up a 1D hydrodynamic model and comparing it to, amongst others, the same hydrodynamic model based on the SRTM DEM. They found the resulting simulated water levels of the SRTM DEM based hydrodynamic model to have a MAD of 0.76m compared to the same levels from the lidar based model. With the proposed procedure, the water levels will be fitted to CryoSat-2 observations. Based on the literature cited the authors assume that fitting the simulated water levels to CryoSat-2 data also means a better fit to real water levels than what can be achieved by setting up the hydrodynamic model based on the SRTM DEM only. Remember also the difficulties of obtaining an estimate of bathymetry from DEMs, whilst the suggested procedure does not require any knowledge of bathymetry. References: Jarihani, A. A., Callow, J. N., McVicar, T. R., Van Niel, T. G. and Larsen, J. R.: Satellite-derived Digital Elevation Model (DEM) selection, preparation and correction for hydrodynamic modelling in large, low-gradient and data-sparse catchments, J. Hydrol., 524, 489–506, doi:10.1016/j.jhydrol.2015.02.049, 2015. Md Ali, A., Solomatine, D. P. and Di Baldassarre, G.: Assessing the impact of different sources of topographic data on 1-D hydraulic modelling of floods, Hydrol. Earth Syst. Sci., 19, 631–643, doi:10.5194/hess-19-631-2015, 2015. |
|---|---|
| changes | p. 14, line 14 – 22: Reformulated the mentioned statement, and discussing the above references |

| Reviewer's comment | *2) Could the procedure be transferable to other case studies? Could the authors suggest the minimum width to apply it?* |
|---|---|
| Authors' response | Yes, the authors expect that this procedure can be transferred to other case studies. A minimum river width however seems to be hard to define, as the ability of satellite altimeters to reliably measure water level in (narrow) rivers depends (besides the actual instrument and processing) not only on the river width, but also on the topography of the river valley – see for example what is discussed in connection with the results shown in Figure 5 and the article by Dehecq et al., 2013 discussed in section 4.1 (p. 11, line 22 – 25) |
| changes | p. 1, line 25: added the transferability to the abstract
p. 15, line 3 – 5: added some words on the transferability to the conclusion |

*TECHNICAL CORRECTIONS:*

| Reviewer's comment | *Please, remove capital letter after the colon.* |
|---|---|
| Authors' response | done |
| changes | throughout the manuscript |

| Reviewer's comment | *Page 3, Line 19: "Mike 11 software": a previous citation of the hydraulic model MIKE 11 used for the analysis is necessary. Please specify if it is a hydrological or hydraulic model and add some references.* |
|---|---|
| Authors' response |  |

| changes | p. 4, line 25 (previously p. 3, line 19): Added reference to MIKE 11 reference manual |
|---|---|
| | p. 7, line 30 – p. 8, line 4: added details on the used hydrodynamic model |

| Reviewer's comment | Table 1: why 27 cross sections? The Envisat tracks are 13 as reported in the pages 8 Line 15. |
|---|---|
| Authors' response | Yes, there exist only 13 virtual stations along the Assam valley. Angles for cross sections lacking neighbouring virtual stations were linearly interpolated between the next cross sections. |
| | We hope we made this more clear together with what is mentioned in the reply to comment 3b) |
| changes | p. 10, line 25 – 26: added a sentence mentioning the above |

Review by anonymous referee #2

| Reviewer's comment | *This paper describes the application of remotely sensed altimetry data from the CryoSat-2 satellite to large scale hydraulic modelling, using the Brahmaputra Basin as an example. While the paper is generally well written and clear, there are a few issues related to the focus and balance of the paper that will need addressing.* |
|---|---|
| | *The remote sensing aspects of the study seem very well described, but the description of the hydraulic modelling is relatively weak. In this respect, the novelty of the work lies in the use of the Cryosat-2 data rather than the hydraulic modelling. In fact given the current research in large scale hydraulic modelling the approach used in the paper is overly simple. Moving beyond the "virtual gauge" is of great research interest and I think this study has real value here, particularly with the fusion of drifting orbit and Envisat virtual stations. The filtering using a dynamic Landsat water mask is also of value and overall I think there is sufficient novelty in the work for publication.* |
| | *While there are some issues to address, I do not think further modelling is required. I think most of the issues can be addressed with changes to the core text. There should be better reference to existing large scale hydraulic river modelling and more discussion/openness about the modelling limitations.* |
| Authors' response | We thank the reviewer for constructive and insightful comments and suggestions. We fully agree with the reviewer that the contribution of this paper is the integration of CryoSat-2 data into a hydrodynamic model (and not for the hydrologic-hydrodynamic modelling as such). |
| changes | - |

*Some more specific points that should be addressed:*

| Reviewer's comment | *(1) The work seems to miss some aspects of recent research that I would assume would be relevant to the work. For example no mention is made of studies that use ICESAT – another dataset that has been used for similar hydraulic model calibration. There is also no reference to the relevant work on channel representation in large scale 1d-2d modelling such as that of Neal et al (2015) (and previous studies).* |
|---|---|
| Authors' response | We agree that the referencing may be too narrow in places and will include the suggested content in the revision. (the 1d-2d model discussion is included as part of the response to the next comment) |
| changes | p. 2, line 8 – 11: added general reference to ICESat
p. 3, line 1 – 3 and p. 3, line 12 – 13: references to studies using ICESat altimetry to compare to or calibrate models |

| Reviewer's comment | *(2) Why only use a 1d model when there are plenty examples of this scale of hydraulic model using 1d&2d? Essentially all the floodplain and braided river section details are being lumped into the single triangular cross-section, so I am not sure how valid the representation of the river/floodplain is in the end. It might work as a simple water level response function that can be calibrated (as demonstrated in the paper), but it losses any physically based reality in representing the river and its floodplain, thereby limiting the value to the model for basin/river/floodplain studies. It is possible of course that the hydraulic conditions are such that the detailed representation of the channel is less important, such as found by Trigg et al 2009 on the Amazon. However there is no detail provided to show this is the case, for example what are the Froude numbers for the flow? It has not been demonstrated that the resulting model has value outside of the modelled scenario. I don't think that the model necessarily has to be redone, but I do think its limitations need more discussion.* |
|---|---|

| Authors' response | Focus here is on accurate prediction of water levels and discharge, this is not a flood model. No predictions about flood extent is possible. The main reason for choosing a 1D model was computational efficiency. It is correct that 1d-2d modelling at this scale is feasible, but probabilistic approaches using large ensembles of model runs would pose significant computational challenges. For example, the cross section calibration presented in the article using a genetic algorithm to find optimal parameters requires many runs (in the range of 10 000) of the model. Moreover, running a meaningful 1d-2d model would require accurate topography/bathymetry, which is unavailable for this braided and highly dynamic river system.
The result of the cross section calibration, especially of step 2 where the amplitudes are being fitted, is consistent with the results by Trigg et al. (2009): with the chosen – simplistic – cross section representation we are able to reproduce observed water level dynamics.
The authors however assume that the study is transferable to other rivers as well, given the availability of sufficient altimetry data. Even if a triangular cross section with varying angle will not be able to reproduce observed water level dynamics for all rivers, the same calibration procedure could be applied to other descriptions of cross section geometry. For example the same procedure should also work for the power-function cross section shape described by Neal et al. (2015): Instead of using a triangular cross section with the angle as only parameter, one could use the power function cross section shape and use i) only shape parameter s or also ii) both shape parameter s and bankfull depth h_full as calibration parameters. (assuming that the third parameter to describe the cross section in Neal et al.'s approach, bankfull width w_full, can easily be estimated from remote sensing data)
Concerning the hydraulic plausibility of the results: The Froude numbers range from ~0.1 to ~0.4, as expectable for the given river section. Also, water depths are mostly in a range between 1 and 10 metres. This means that the hydrodynamic model with the performed cross section calibration behaves somewhat as expected (e.g.: subcritical flow), at least not in physically implausible regions. |
|---|---|
| changes | (introduced sub-sections 1.1 to 1.3 in the now extended introduction for clarity)
p. 3, line 31 – p. 4, line 10: Added section 1.3 to discuss the choice of a 1D over a 2D model
p. 13, line 30 – p. 14, line 3: added a few words on the limitations of the chosen synthetic cross section shapes to the discussion, including a reference to the work by Trigg et al. (2009) and Neal et al. (2015)
p. 13, line 26 – 30: Added some values on the discussed hydraulic plausibility of the hydrodynamic model |

| Reviewer's comment | *(3) More discussion is required on the uncertainty in flow produced by the rainfall runoff modelling and how it affects the hydraulic modelling.* |
|---|---|
| Authors' response | The uncertainty in the subcatchments' rainfall-runoff models indeed is quite big (see Table 4, which was added in the revision). However, when aggregated in the larger hydrodynamic model, these uncertainties cancel each other out, producing a better model fit on the basin level than on the subcatchment level. the average cross correlation coefficient of the residuals of the subcatchments' runoff is only 0.16. Include some details and discussion of the uncertainty of the rainfall-runoff models. Potentially including data from the calibration catchments. |
| changes | added new Table 4: Performance criteria for simulated discharge in the calibration subcatchments.
p. 12, line 25 – 32: added some discussion of the uncertainty on subcatchment vs. basin level |

| Reviewer's comment | *(4) There is reference to the dynamic nature of river channel with regards to the water mask, but no discussion of the how important this geomorphology might be to the simple triangle river channel model used.* |
|---|---|

| Authors' response | For this simple 1D model, with its synthetic cross sections, we assume that the change of the river channels does not significantly affect the water level-discharge relationships. Also, Mirza (2003) found rating curves at the Brahmaputra to exhibit fairly constant Q-h relationships over decades, leading to the conclusion that the Brahmaputra River is in "dynamic equilibrium".
Reference: Mirza, M. M. Q.: The Choice of Stage-Discharge Relationship for the Ganges and Brahmaputra Rivers in Bangladesh, Nord. Hydrol., 34(4), 321–342, doi:10.2166/nh.2003.019, 2003.
(see also response to comment 2 and 3a in Data and Methods by anonymous referee #1) |
|---|---|
| changes | p. 8, line 5 – 10: added a few words on this, including the above reference |

| Reviewer's comment | *(5) I am not clear on how the SRTM is actually translated into the triangle river channel. Has the raw SRTM data been processed to remove the vegetation bias? What is actually used for the 1d triangle, the width and depth of the river extracted from the SRTM? If so maybe river width from landsat would be better for the width and estimate of depth from geomorphological relationships (Leopold, and Maddock, 1953) would be better? What size are these calibrated triangles. Do they bear any resemblance to the real river sections?* |
|---|---|
| Authors' response | What is referred to as "reference" cross sections in the paper was visually extracted from satellite imagery and the SRTM DEM in a consulting project preparing the Ganges-Brahmaputra hydrologic model used in this paper. The real river cross sections (of this multi-channelled river) will of course be very different from these simplistic cross sections. However, this is not so important in a 1D model, as long as the relationship A = A(h) and P = P(h) are realistic, i.e. we need to get the relationship between flow cross sectional area and wetted parameter right. |
| changes | p. 9, line 14 – 20: added explanations on the "reference" cross sections and their limitations
p. 13, line 3 – 4: added reference to the explanation of "reference" cross sections above |

| Reviewer's comment | *(6) Manning's is mentioned but no values given. Given its direct control on water levels and it should have some link to expected values it should not be omitted. Given the crude nature of the cross-sections and the fact that Manning's will compensate for lots of missing processes in this regard, I am not sure the calibrated Manning's values will bear resemblance to what might be expected for such a river.* |
|---|---|
| Authors' response | The resulting value is a Manning's n value of 0.029 in SI units, which is considered plausible (compare for example
http://www.fsl.orst.edu/geowater/FX3/help/8_Hydraulic_Reference/Mannings_n_Tables.htm)
(same as comment 2 to Result section by anonymous referee #1) |
| changes | p. 12, line 10 – 11: added the above |

[revised manuscript text omitted]

---

## Author Response (AR2)

| Reviewer's comment | I appreciated the description of the hydraulic and rainfall-runoff model and the added comments. All is more clear and balanced. I would like to know: which are the parameters necessary to run the rainfall-runoff model (maybe it could be added in the manuscript)? why the authors did not use the two gauged stations of Hardinge Bridge and Bahadurabad station for the calibration of the parameters? |
|---|---|
| Authors' response | Added a sentence explaining what elements of the model are controlled by the calibrated parameters. For more detail, like a full listing of parameters and their names in NAM a reference to the MIKE 11 Reference Manual was added. |
| changes | p. 8, line 28 – p. 9, line 4 |

| Reviewer's comment | Why the authors write the Manning value in SI units? It can be specified in s/m^(1/3). |
|---|---|
| Authors' response | changed |
| changes | p.12, line 12 |

| Reviewer's comment | I believe that Figure 1 can be improved by adding some references that the authors cite along the text (Tibetan Plateau, Assam Valley, Himalayan Mountains). Moreover, could be interesting also to see the location of the 2 gauged stations inside the Brahmaputra basin. Concerning the problem to have the station of Bahadurabad outside of the basin, GIS could be forced with the river line, in the way to have the edge of the basin coherent with the river. If the authors derived the river shape from SRTM, how they obtain this uncoherent shape of the basin? On the SCALGOLIVE website, I checked the limits of the Brahmaputra river basin and it seems right. I suggest the authors to check the polygon. |
|---|---|
| Authors' response | Figure 1 was improved by adding the mentioned features (Tibetan Plateau etc.) in the caption. Also, the gauge locations now are included. Concerning the basin lineout: The deviation originates from the very low slope in this region, making extraction of exact flow direction maps from SRTM hard. (some inconsistencies can also be seen when checking watersheds based on SRTM on the SCALGOLIVE website). To avoid confusion, we manually corrected the basin outline to fall together with the river line as suggested by the reviewer. |
| changes | changed Figure 1 and extended its caption |

| Reviewer's comment | The part regarding the cross-section calibration is improved, even if the reader can be a little bit confused by the different number of cross sections described in the text (13 virtual station) and in the Table 1 (27 for ENVISAT and 24 for Cryosat-2). It should be appreciate a comment of Table 1 for clarifying the different number. |
|---|---|
| Authors' response | We added a sentence explaining the difference between no. of virtual stations and no. of cross sections after the section explaining the interpolation of cross sectional parameters across regions without virtual station data (referring to Table 1). |
| changes | p. 10, line 27 – 28 |

[revised manuscript text omitted]